# Continual Learning: Less Forgetting, More OOD Generalization via Adaptive Contrastive Replay

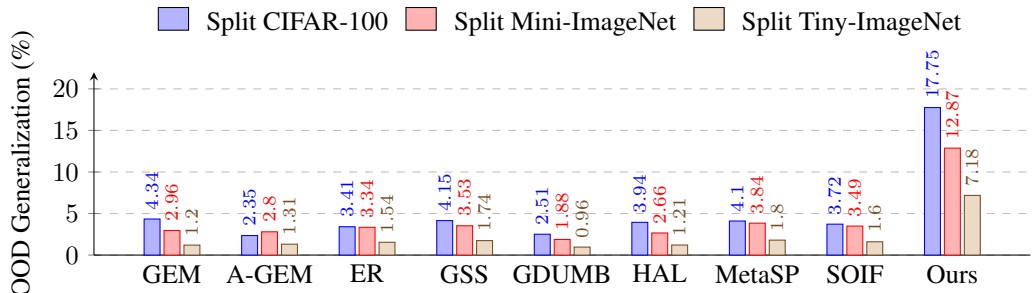

Figure 1: **Evaluating Out-of-Distribution (OOD) Generalization Capability:** The performance of state-of-the-art rehearsal-based methods on the Split CIFAR-100, Split Mini-ImageNet, and Split Tiny-ImageNet datasets significantly drops on OOD samples, highlighting their lack of generalization. In this paper, we address this challenge by proposing a method that consistently outperforms existing approaches across all datasets.

## Abstract

Machine learning models often suffer from catastrophic forgetting of previously learned knowledge when learning new classes. Various methods have been proposed to mitigate this issue. However, rehearsal-based learning, which retains samples from previous classes, typically achieves good performance but tends to memorize specific instances, struggling with Out-of-Distribution (OOD) generalization. This often leads to high forgetting rates and poor generalization. Surprisingly, the OOD generalization capabilities of these methods have been largely unexplored. In this paper, we highlight this issue and propose a simple yet effective strategy inspired by contrastive learning and data-centric principles to address it. We introduce **Adaptive Contrastive Replay (ACR)**, a method that employs dual optimization to simultaneously train both the encoder and the classifier. ACR adaptively populates the replay buffer with misclassified samples while ensuring a balanced representation of classes and tasks. By refining the decision boundary in this way, ACR achieves a balance between stability and plasticity. Our method significantly outperforms previous approaches in terms of OOD generalization, achieving an improvement of 13.41% on Split CIFAR-100, 9.91% on Split Mini-ImageNet, and 5.98% on Split Tiny-ImageNet.[1]

## 1 Introduction

Continual Learning (CL), the gradual acquisition of new concepts (classes or tasks) without forsaking previous ones, stands as a pivotal capability in machine learning. Various methodologies, including regularization techniques Kirkpatrick et al. (2017); Chaudhry et al. (2018a), architecture-based approaches Mallya & Lazebnik (2018); Hung et al. (2019), and rehearsal-based strategies

---

[1]Code is available at: https://anonymous.4open.science/r/ACR-3E86

Lopez-Paz & Ranzato (2017); Chaudhry et al. (2018b; 2019); Aljundi et al. (2019); Prabhu et al. (2020); Chaudhry et al. (2021); Sun et al. (2022; 2023), have been explored. However, the central challenge remains in striking a balance between assimilating new concepts (plasticity) and preserving existing knowledge (stability) Kim et al. (2023); Kim & Han (2023).

Current rehearsal-based methods achieve good performance but often neglect the necessity for acceptable Out-of-Distribution (OOD) generalization. This means they are mostly memorized instead of learning from previous classes Verwimp et al. (2021); Zhang et al. (2022). The state-of-the-art CL approaches exhibit significant performance drops when encountering OOD samples where the sample is altered with a slight covariant shift. Yet, they must be capable of generalizing (refer to Figure 1).

The aspect of OOD generalization in CL methods has largely been overlooked until now. While previous studies have concentrated on mitigating overfitting—a factor that can influence generalizability—they have often overlooked OOD generalization Bonicelli et al. (2022); Yu et al. (2022). However, it is critical to acknowledge that merely memorizing previous tasks is not sufficient to prevent forgetting them. Effective methods must not only retain information about prior tasks or classes but also avoid overfitting specific samples and neglecting OOD generalization.

Note that the previous methods aim to retain information from previous classes/tasks to avoid forgetting, albeit through memorization. Preventing forgetting by relying on memorization can be problematic. We can categorize memorization into *good memorization* and *bad memorization*. *Good memorization* ensures that past information is retained without leading to overfitting. In contrast, *bad memorization* focuses on remembering previous tasks without considering the risk of overfitting Bartlett et al. (2020); Li et al. (2021); Tirumala et al. (2022); Wei et al. (2024). Current approaches often suffer from *bad memorization*. Moreover, as current methods report average accuracy across classes, they can hide specific problems, such as significantly poor performance and forgetting of earlier tasks (low stability), and high accuracy on new tasks (high plasticity) (refer to Section 4.2, Figure 3), which also results in poor generalization.

Our findings reveal that existing rehearsal-based methods typically focus on sample contribution when updating the buffer. This often leads to class and task imbalances within the buffer (refer to Section 4.2, Table 3), resulting in a long-tail memory distribution that disrupts decision boundaries and causes poor generalization Cui et al. (2019); Samuel & Chechik (2021); Shi et al. (2023). Moreover, these methods generally exhibit long running times (Appendix A.2, Table 5) and require high GPU memory (Appendix A.2, Figure 5). These limitations restrict their applicability in resource-constrained environments.

To address these issues, we first demonstrate the inadequate OOD generalization capabilities of existing methods, including Lopez-Paz & Ranzato (2017); Chaudhry et al. (2018b; 2019); Aljundi et al. (2019); Prabhu et al. (2020); Chaudhry et al. (2021); Sun et al. (2022; 2023), using the methodology described in Hendrycks & Dietterich (2019) (Figure 1). Building on this analysis and inspired by the proxy-based contrastive learning outlined in Yao et al. (2022) as well as data-centric approaches described in Toneva et al. (2018); Swayamdipta et al. (2020), we introduce **A**daptive **C**ontrastive **R**eplay (**ACR**). ACR not only outperforms existing methods on both i.i.d. and OOD samples but also reduces resource requirements. In the supplementary material, we comprehensively and in detail discussed the related works (Appendix A.1).

The main contributions of this paper are: (1) To our knowledge, this is the first work to demonstrate that the performance of rehearsal-based CL methods significantly degrades under distributional shift conditions where the i.i.d. assumption does not hold. (2) We leverage contrastive learning to introduce a dual optimization objective while populating the buffer with misclassified samples (except outliers). This approach simultaneously optimizes both the encoder and classifier using contrastive loss and ensures that the buffer is populated with boundary samples. (3) We maintain a balanced distribution of classes and tasks within the buffer, ensuring that all categories are adequately represented. (4) Our method achieves a better-balanced trade-off between stability and plasticity compared to existing approaches, leading to more robust performance. (5) Our approach is both simple and effective, requiring fewer resources and less running time compared to higher-performing methods, making it more practical for real-world applications.

## 2 PRELIMINARIES

Consider a model with an encoder $f_\theta$ and a classifier $g_\phi$ parameterized by $\theta$ and $\phi$ respectively, which are trained incrementally on a series of tasks $\{1, 2, \ldots, T\}$. At a specific time $t$, the model processes the task $\mathcal{D}_t = \{(x_i^t, y_i^t), \mathcal{C}^t\}_{i=1}^{N^t}$, where $x_i^t$ represents the input data, $y_i^t$ the corresponding label, $N^t$ the number of samples in $\mathcal{D}_t$, and $\mathcal{C}^t$ indicates the classes specific to task $\mathcal{D}_t$. The buffer $\mathcal{B}$ is used to store a subset of past examples to mitigate catastrophic forgetting while having a limited capacity, denoted as $B_{\text{size}}$. For each task $\mathcal{D}_t$, the model's objective is to learn the new task while retaining knowledge from previous tasks. The model parameters are updated from $\theta_{t-1}, \phi_{t-1}$ to $\theta_t, \phi_t$ by minimizing the loss function over the combined dataset of the current task and the buffer:

$$\theta_t, \phi_t = \arg \min_{\theta_{t-1}, \phi_{t-1}} \left[ \mathcal{L}(\mathcal{D}_t, \mathcal{B}, f_{\theta_{t-1}}, g_{\phi_{t-1}}) \right] \tag{1}$$

where $\mathcal{L}$ represents the loss function.

## 3 PROPOSED METHOD: ADAPTIVE CONTRASTIVE REPLAY (ACR)

### 3.1 MOTIVATION AND INTUITION:

As previously mentioned, rehearsal-based CL methods often face challenges like overfitting Bonicelli et al. (2022) and class/task representation imbalances, leading to long-tail memory effects Cui et al. (2019). These issues result in less distinct representations, poor OOD generalization, higher forgetting rates, and an imbalanced stability-plasticity trade-off due to blurred decision boundaries. Additionally, their optimization objectives (e.g., GEM Lopez-Paz & Ranzato (2017) to prevent loss increases based on old tasks' samples) and sample selection strategies (e.g., GSS Aljundi et al. (2019) to ensure gradient space diversity, MetaSP Sun et al. (2022), and SOIF Sun et al. (2023) to use influence functions) increase GPU usage and training times.

To address these issues, we propose a novel approach called Adaptive Contrastive Replay (ACR), which adaptively updates the replay buffer while employing a proxy-based contrastive loss. The proxy-based contrastive loss not only aids in optimizing the encoder to achieve distinct representations but also optimizes the classifier simultaneously, resulting in faster operations and lower memory usage compared to traditional contrastive loss methods. Additionally, to identify boundary samples during training, we use the model's confidence scores, which allows us to select informative samples with minimal computational overhead. When updating the buffer, we ensure that the number of samples per class and task is balanced, thus avoiding the long-tail memory effect. By combining proxy-based contrastive loss with our adaptive buffer update strategy, ACR achieves superior representation learning, reduced memory usage and computation time, improved OOD generalization, and a better balance between stability and plasticity.

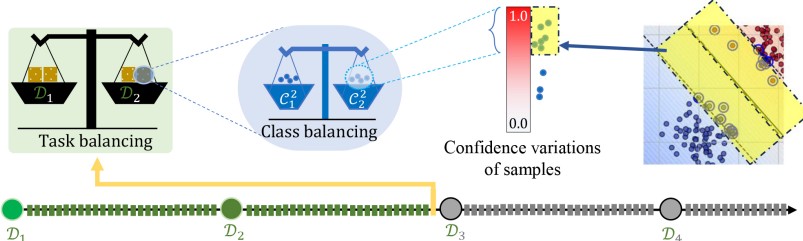

Figure 2: Illustration of the buffer update policy in our method (ACR). After training each task, the buffer is updated with the most challenging samples, identified by high confidence variation, while maintaining class and task balance.

### 3.2 CONFIDENCE-GUIDED SAMPLE SELECTION AND PROXY-BASED CONTRASTIVE LEARNING INTEGRATION

As shown in Figure 2, not all samples contribute equally to CL tasks. To address this, ACR prioritizes the most informative samples by measuring confidence variance across epochs, selecting those near the decision boundaries. This ensures that the model retains critical boundary samples,

enhancing stability and reducing forgetting. Additionally, ACR integrates proxy-based contrastive learning, which improves feature representation by maintaining class separation while minimizing computational overhead.

**Confidence Variance:** The confidence of the model in its predictions is defined as the probability assigned to the target class $y_i^t$ for sample $x_i^t$ during task $\mathcal{D}_t$. Formally, the confidence score is expressed as:

$$\Gamma(x_i^t, y_i^t) = \mathcal{P}(y_i^t \mid g_\phi(f_\theta(x_i^t), \mathcal{W})) \tag{2}$$

where $f_\theta$ is the encoder that maps the input $x_i^t$ into a latent space, and $g_\phi$ is the classifier that assigns class probabilities based on the encoded features. During training, misclassified samples fall into two categories: boundary samples and outlier samples. Outlier samples consistently exhibit low confidence, whereas boundary samples display fluctuating confidence levels, reflecting uncertainty in their classification Toneva et al. (2018); Swayamdipta et al. (2020). This fluctuation can be captured through the variance of the confidence score over the first $E$ epochs of training:

$$\sigma^2(x_i^t, y_i^t) = \frac{1}{E} \sum_{e=1}^{E} \left( \Gamma_e(x_i^t, y_i^t) - \overline{\Gamma}(x_i^t, y_i^t) \right)^2 \tag{3}$$

where $\Gamma_e(x_i^t, y_i^t)$ represents the confidence at epoch $e$, and $\overline{\Gamma}(x_i^t, y_i^t)$ is the average confidence over $E$ epochs. Higher values of $\sigma^2(x_i^t, y_i^t)$ suggest that the sample lies near a decision boundary where the classification of the model is unstable.

**Proxy Based Contrastive Loss:** Softmax loss builds positive and negative pairs using proxy-to-sample relationships Sun et al. (2023), where proxies represent sub-datasets, making it robust against noise and outliers Yao et al. (2022). The anchor is a sample from the batch. In contrast, supervised contrastive loss forms positive pairs from samples within the same class, focusing on sample-to-sample similarity in the batch Mai et al. (2021). The key difference is that contrastive methods focus on sample-to-sample interactions, while proxy-based methods use proxies for faster, safer convergence but may miss some semantic relationships. Both methods have limitations: contrastive loss poses challenges in stability and training complexity when sample availability is limited, while softmax loss struggles with class imbalance and misclassification of older class samples as newer ones.

To address these challenges, we propose a proxy-based contrastive learning approach that replaces traditional sample-to-sample comparisons with proxy-to-sample relations, effectively reducing the issues related to positive pair alignment. Our method, the proxy-based contrastive loss, links each proxy to all data samples in a batch, thereby substantially expanding the pool of negative samples. In this approach, each proxy acts as an anchor, and we leverage all corresponding proxy-to-sample interactions.

For each task $\mathcal{D}_t$, current samples $(x_i^t, y_i^t)^b$ and buffer samples $(x_i^{\mathcal{B}}, y_i^{\mathcal{B}})^b$, along with their augmented versions, are combined to form the training batch $(\mathbf{x}, y)$ for robust optimization. Then the model is optimized using this training batch and the proxy-based contrastive loss function is defined by the following equation:

$$\mathcal{L}_{PCL} = -\frac{1}{N} \sum_{i=1}^{N} \log \frac{\exp\left(f_\theta(\mathbf{x}) \cdot \mathcal{W}_y / \tau\right)}{\sum_{L \in \{\mathbf{C}\}} \exp\left(f_\theta(\mathbf{x}) \cdot \mathcal{W}_L / \tau\right)} \tag{4}$$

where $f_\theta(\mathbf{x})$ represents the latent embeddings of $\mathbf{x}$, $\mathcal{W}_y$ is the class proxy (weight) for class $y$, $\tau$ is the temperature parameter, and $\{\mathbf{C}\}$ is the set of class labels in the batch. This loss function aligns the embeddings of each sample $\mathbf{x}$ with the weight of the target class $\mathcal{W}_y$ while distancing them from other class proxies.

As the confidence variance of a sample increases, its distance from the decision boundary, $d(\mathbf{x}_i^t)$, decreases, making it more sensitive to model changes:

$$\frac{\partial \Gamma(\mathbf{x}_i^t, y_i^t)}{\partial d(\mathbf{x}_i^t)} > 0 \tag{5}$$

This proxy-based loss ensures tighter alignment of samples with their class proxies, refining the decision boundaries and minimizing errors in future tasks, leading to improved generalization.

The full process of ACR is summarized in Algorithm 1, which illustrates how confidence-guided sample selection, proxy-based contrastive learning, and buffer management are combined to improve continual learning performance.

---

**Algorithm 1** Adaptive Contrastive Replay (ACR)

---

1: **Input:** Tasks $\{\mathcal{D}_1, \ldots, \mathcal{D}_T\}$, encoder $f_\theta$, classifier $g_\phi$, buffer $\mathcal{B}$, confidence variance threshold $\tau_\sigma$, temperature $\tau$, ACR hyperparameter $E$, epochs number $m$, task and buffer batch size $b$
2: **Initialize:** Empty replay buffer $\mathcal{B}$, parameters $\theta$, $\phi$
3: **for** $t = 1$ to $T$ **do**
4:     **for** epoch $e = 1$ to $m$ **do**
5:         **for** $(x_i^t, y_i^t)^b \sim \mathcal{D}_t$ **do**
6:             $(x_i^\mathcal{B}, y_i^\mathcal{B})^b \leftarrow \text{RandomRetrieval}(\mathcal{B})$
7:             $(\tilde{x}_i^t, y_i^t)^b \leftarrow \text{DataAugmentation}(x_i^t, y_i^t)^b$
8:             $(\tilde{x}_i^\mathcal{B}, y_i^\mathcal{B})^b \leftarrow \text{DataAugmentation}(x_i^\mathcal{B}, y_i^\mathcal{B})^b$
9:             $(\mathbf{x}, y) \leftarrow \text{Concatenation}\{(x_i^t, y_i^t)^b, (x_i^\mathcal{B}, y_i^\mathcal{B})^b, (\tilde{x}_i^t, y_i^t)^b, (\tilde{x}_i^\mathcal{B}, y_i^\mathcal{B})^b\}$
10:             $\mathbf{z} \leftarrow f_\theta(\mathbf{x})$
11:             **if** $e < E$ **then**
12:                 $\Gamma_e(x_i^t, y_i^t)^b \leftarrow \mathcal{P}((y_i^t)^b \mid g_\phi(\mathbf{z}[:b], \mathcal{W}))$           $\triangleright$ $\Gamma$: confidences of $(x_i^t)^b$.
13:             **end if**
14:             $\mathcal{L}_{PCL} = -\frac{1}{N} \sum_{i=1}^{N} \log \frac{\exp(\mathbf{z} \cdot \mathcal{W}_y / \tau)}{\sum_{L \in \{\mathbf{C}\}} \exp(\mathbf{z} \cdot \mathcal{W}_L / \tau)}$
15:             Update $\theta$ and $\phi$
16:         **end for**
17:     **end for**
18:     $\sigma^2(x^t, y^t) = \frac{1}{E} \sum_{e=1}^{E} \left( \Gamma_e(x^t, y^t) - \overline{\Gamma}(x^t, y^t) \right)^2$
19:     $\mathcal{B} \leftarrow \mathcal{B} \setminus \left\{ x_{i,c}^{<t} \mid \sigma^2(x_{i,c}^{<t}, y_{i,c}^{<t}) \text{ ranks among the lowest} \right\}$
20:     $\mathcal{B} \leftarrow \mathcal{B} \cup \left\{ x_{i,c}^{t} \mid \sigma^2(x_{i,c}^{t}, y_{i,c}^{t}) > \tau_\sigma \right\}$
21: **end for**
22: **Output:** Updated model parameters $\theta$, $\phi$, and replay buffer $\mathcal{B}$

---

### 3.3 ADAPTIVE REPLAY BUFFER MANAGEMENT

Selecting high-variance samples near decision boundaries enhances contrastive learning by refining these boundaries, improving OOD generalization, and preventing overfitting. This approach maintains a balance between stability (retaining past knowledge) and plasticity (adapting to new tasks), reducing catastrophic forgetting. ACR manages these critical samples in its replay buffer, ensuring balanced representation across classes and tasks, preventing learning imbalances, and improving overall model performance.

**Task- and Class-Aware Buffer Structure:** Unlike conventional rehearsal-based methods that neglect balanced class/task distribution in the buffer, ACR organizes its buffer $\mathcal{B}$ into task-specific and class-specific partitions:

$$\mathcal{B} = \bigcup_{t=1}^{T} \mathcal{B}^t = \bigcup_{t=1}^{T} \bigcup_{c \in \mathcal{C}^t} \mathcal{B}_c^t \tag{6}$$

where $\mathcal{C}^t$ denotes the set of classes within task $\mathcal{D}_t$, $\mathcal{B}^t$ represents the buffer allocated for task $\mathcal{D}_t$, and $\mathcal{B}_c^t$ indicates the buffer designated for class $c$ within task $\mathcal{D}_t$. For each task at time $t$, the buffer allocates $\frac{B_{\text{size}}}{t}$ space, and for each class within task $\mathcal{D}_t$, it allocates $\frac{B_{\text{size}}}{t \times |\mathcal{C}^t|}$ space, where $|\mathcal{C}^t|$ shows the number of class in task $\mathcal{D}_t$. This structure ensures that the replay buffer maintains a balanced distribution of classes and tasks, preventing the model from overfitting to certain dominant classes while neglecting others.

In the buffer, tasks are arranged sequentially by order of arrival. Additionally, each task's classes are stored separately within the buffer. Within each class, samples are ordered by decreasing $\sigma^2$. Consequently, as the buffer index increases, the task number also increases while the $\sigma^2$ of the samples for each class decreases. This structure of the buffer eliminates the need to store the task label and the $\sigma^2$ of each sample. To manage and maintain the class samples with the highest $\sigma^2$, we just store samples and corresponding labels in the buffer like traditional methods.

**Confidence-Based Buffer Update and Pruning:** At the end of each task, the buffer is updated with samples that exhibit the highest confidence variance within each class. This selective updating ensures that only the most uncertain and informative samples are retained. The buffer update rule is given by:

$$\mathcal{B} \leftarrow \mathcal{B} \cup \left\{ x^t_{i,c} \mid \sigma^2(x^t_{i,c}, y^t_{i,c}) > \tau_\sigma \right\} \tag{7}$$

where the $\tau_\sigma$ is adjusted so that the number of selected samples for each class equals $\frac{B_{\text{size}}}{t \times |\mathcal{C}^t|}$, ensuring balanced classes.

To manage memory constraints, ACR implements a pruning mechanism whereby it removes the samples with the lowest variance from each class before updating the buffer with a new task, $\mathcal{D}_t$. Specifically, for a class $c$ from the set of previous classes $\mathcal{C}^{t-1}$, the buffer space of $\frac{B_{\text{size}}}{(t-1) \times |\mathcal{C}^{t-1}|} - \frac{B_{\text{size}}}{t \times |\mathcal{C}^{t-1}|}$ is cleared:

$$\mathcal{B} \leftarrow \mathcal{B} \setminus \left\{ x^{<t}_{i,c} \mid \sigma^2(x^{<t}_{i,c}, y^{<t}_{i,c}) \text{ ranks among the lowest} \right\} \tag{8}$$

This selective removal ensures that each update retains the most informative samples, which are crucial for maintaining accurate decision boundaries.

# 4 EXPERIMENTS

## 4.1 SETUP

For our experiments, we extended the Mammoth framework as described in Buzzega et al. (2020); Boschini et al. (2022) to ensure a fair setup for comparing CL methods. All experiments are conducted using Quadro RTX 8000 equipped with CUDA Version 10.2.

**Datasets.** Our experiments utilized three benchmarks. The first, Split CIFAR-100, consists of 100 classes split into 10 tasks, each with 10 classes and $32 \times 32$ pixels images Boschini et al. (2022). The second, Split Mini-ImageNet, divides 100 classes into 5 tasks of 20 classes each, with similarly resized images Sun et al. (2022). The third, Split Tiny-ImageNet, includes 200 classes split into 10 tasks of 20 classes each, with images resized to $32 \times 32$ pixels Buzzega et al. (2020). Each dataset has 500 training samples per class, with test samples numbering 100 for Split CIFAR-100 and Mini-ImageNet, and 50 for Split Tiny-ImageNet.

To generate OOD images, we follow the methodology outlined in Hendrycks & Dietterich (2019). This approach involves applying common corruptions such as Gaussian Noise, Shot Noise, Impulse Noise, Defocus Blur, Motion Blur, Zoom Blur, Snow, Fog, Elastic Transform, Pixelate, and JPEG Compression.

**Metrics.** We employ two primary metrics to evaluate the performance of CL methods: Average Accuracy (ACC) and Backward Transfer (BWT), as outlined in Lopez-Paz & Ranzato (2017). The ACC metric assesses overall performance by calculating the average accuracy across all tasks after the model has completed training. Conversely, BWT measures the degree of knowledge loss over time by measuring the average forgetting across all previous tasks once training on all tasks has concluded. These metrics are mathematically defined as follows:

$$\text{ACC} = \frac{1}{T} \sum_{j=1}^{T} \alpha_{T,j}, \quad \text{BWT} = \frac{1}{T-1} \sum_{j=1}^{T-1} \beta_{T,j} \tag{9}$$

Here, $\alpha_{i,j}$ represents the accuracy on the test set held out for task $j$ after the network has been trained on task $i$. $\beta_{i,j}$ is calculated as $(\alpha_{i,j} - \alpha_{j,j})$, signifying the decrease in model performance on task $j$ due to training on subsequent tasks. $T$ shows the number of tasks. We evaluate these metrics under class incremental conditions, as discussed in previous studies Boschini et al. (2022).

**Baselines.** We compared our method against eight rehearsal-based approaches, including GEM Lopez-Paz & Ranzato (2017), A-GEM Chaudhry et al. (2018b), ER Chaudhry et al. (2019), GSS Aljundi et al. (2019), GDUMB Prabhu et al. (2020), HAL Chaudhry et al. (2021), MetaSP Sun et al. (2022), and SOIF Sun et al. (2023).

Table 1: Results of class-incremental learning (Average Accuracy (ACC), higher is better: ↑), averaged over 5 runs, compare various methods across three datasets under different memory constraints in both i.i.d. and OOD scenarios. 'Mean' columns within each dataset show averages across buffer sizes, while the 'Mean' column next to the datasets indicates the overall average for each method across all datasets. The best results are bolded, and the second best are underlined.

| Task | | Split CIFAR-100 | | | | Split Mini-ImageNet | | | | Split Tiny-ImageNet | | | | Mean |
|---|---|---|---|---|---|---|---|---|---|---|---|---|---|---|
| Buffer | | 500 | 1000 | 2000 | Mean | 500 | 1000 | 2000 | Mean | 500 | 1000 | 2000 | Mean | |
| GEM | i.i.d. | 21.33 | 31.91 | 36.33 | 29.86 | 18.04 | 19.02 | 18.99 | 18.68 | 6.07 | 6.92 | 7.08 | 6.69 | 18.41 |
| | OOD | 3.36 | 4.69 | 4.98 | 4.34 | 2.97 | 2.97 | 2.94 | 2.96 | 1.18 | 1.20 | 1.22 | 1.20 | 2.83 |
| A-GEM | i.i.d. | 9.32 | 9.23 | 9.25 | 9.27 | 14.65 | 14.69 | 14.74 | 14.69 | 6.30 | 6.30 | 6.23 | 6.28 | 10.08 |
| | OOD | 2.31 | 2.39 | 2.36 | 2.35 | 2.72 | 2.82 | 2.86 | 2.80 | 1.30 | 1.33 | 1.29 | 1.31 | 2.15 |
| ER | i.i.d. | 14.99 | 18.58 | 24.11 | 19.23 | 16.64 | 18.45 | 21.17 | 18.75 | 8.20 | 9.57 | 11.78 | 9.85 | 15.94 |
| | OOD | 2.97 | 3.36 | 3.91 | 3.41 | 3.12 | 3.33 | 3.58 | 3.34 | 1.38 | 1.49 | 1.76 | 1.54 | 2.76 |
| GSS | i.i.d. | 22.23 | 28.84 | 35.27 | 28.78 | 18.48 | 21.91 | 28.16 | 22.85 | 9.23 | 11.45 | 15.52 | 12.07 | 21.23 |
| | OOD | 3.45 | 4.13 | 4.87 | 4.15 | 3.19 | 3.43 | 3.96 | 3.53 | 1.49 | 1.70 | 2.03 | 1.74 | 3.14 |
| GDUMB | i.i.d. | 10.36 | 16.35 | 25.47 | 17.39 | 7.09 | 10.15 | 14.33 | 10.52 | 3.59 | 5.15 | 7.53 | 5.42 | 11.11 |
| | OOD | 1.76 | 2.45 | 3.33 | 2.51 | 1.49 | 1.83 | 2.33 | 1.88 | 0.79 | 0.93 | 1.17 | 0.96 | 1.78 |
| HAL | i.i.d. | 10.23 | 12.62 | 16.68 | 13.18 | 5.95 | 7.10 | 8.59 | 7.21 | 2.45 | 2.58 | 2.91 | 2.65 | 7.68 |
| | OOD | 3.44 | 3.89 | 4.50 | 3.94 | 2.47 | 2.59 | 2.66 | 2.57 | 1.15 | 1.20 | 1.29 | 1.21 | 2.60 |
| MetaSP | i.i.d. | 18.82 | 25.15 | 32.87 | 25.61 | 19.92 | 23.67 | 28.92 | 24.17 | 9.49 | 12.45 | 16.14 | 12.69 | 20.83 |
| | OOD | 3.40 | 4.04 | 4.86 | 4.10 | 3.40 | 3.84 | 4.28 | 3.84 | 1.51 | 1.80 | 2.09 | 1.80 | 3.25 |
| SOIF | i.i.d. | 18.85 | 26.19 | 25.36 | 23.47 | 18.75 | 22.85 | 23.46 | 21.69 | 9.10 | 11.66 | 11.58 | 10.78 | 18.64 |
| | OOD | 3.18 | 3.96 | 4.03 | 3.72 | 3.23 | 3.56 | 3.68 | 3.49 | 1.46 | 1.69 | 1.65 | 1.60 | 2.94 |
| **Ours** | i.i.d. | **29.27** | **36.06** | **42.24** | **35.86** | **23.88** | **28.49** | **32.72** | **28.36** | **12.47** | **16.40** | **21.19** | **16.69** | **26.97** |
| | OOD | **14.81** | **17.86** | **20.57** | **17.75** | **11.10** | **12.90** | **14.62** | **12.87** | **5.74** | **7.06** | **8.73** | **7.18** | **12.60** |

**Implementation Details.** For the implementation and hyperparameters of the baselines, we followed the Mammoth framework Buzzega et al. (2020); Boschini et al. (2022), MetaSP Sun et al. (2022), and SOIF Sun et al. (2023), using ResNet18 He et al. (2016) as the backbone model, trained from scratch. We ensured fairness by averaging results over five runs with fixed seeds (0 to 4). Both the current and memory batch size were set to 32. The hyperparameter $E$ in our method was empirically set to 5 for all three datasets by default, with its sensitivity analyzed in Appendix A.2 and Figure 4. We used the SGD optimizer for 50 epochs per task, with learning rates of 0.1 for Split CIFAR-100 and 0.03 for both Split Mini-ImageNet and Split Tiny-ImageNet, applying standard augmentations, including random cropping and random horizontal flipping.

## 4.2 MAIN RESULTS

In Tables 1, and 2, we present the quantitative results of all compared methods, including the proposed ACR, in class-incremental settings for both i.i.d. and OOD scenarios. Table 1 details the Average Accuracy (ACC), and Table 2 reports the Backward Transfer (BWT).

As shown in Table 1, our method consistently outperformed all others across both i.i.d. and OOD scenarios in all three datasets. Specifically, for Split CIFAR-100, we achieved a 6.0% higher average ACC than the second-best result in the i.i.d. scenario and a 13.41% margin in the OOD scenario. For Split Mini-ImageNet, our method surpassed competitors by 4.19% in the i.i.d. scenario and 9.03% in the OOD scenario. On Split Tiny-ImageNet, we outperformed the second-best by 4.0% in the i.i.d. scenario and 5.38% in the OOD scenario. Overall, across all datasets, our method maintained an average margin of 5.74% over the second-best in the i.i.d. scenario and 9.35% in the OOD scenario.

In Table 2, we show that for Split CIFAR-100, our method exceeded the second-best by an average BWT of 16.2% in the i.i.d. scenario and 1.61% in the OOD scenario. For Split Mini-ImageNet, our method ranked second in the i.i.d. scenario. However, a closer analysis is warranted. In this case, the HAL method achieves the best average BWT in the i.i.d. scenario. Nonetheless, as Table 1 shows, HAL has the worst average ACC (7.21%), indicating that it failed to acquire sufficient knowledge to retain, much less forget, which raises concerns about its overall learning capability. Therefore, a fair comparison should consider both BWT and ACC, as high BWT alone is insufficient if the ACC is low. To this end, we should consider BWT alongside ACC.

In this regard, in Table 1, our method achieves an average ACC of 28.36% in the i.i.d. scenario on this dataset. In contrast, the next best methods—MetaSP (24.17%), GSS (22.85%), SOIF (21.69%),

Table 2: Results of class-incremental learning with Backward Transfer (BWT) metrics, favoring less negative values, averaged over five runs. The analysis covers various methods across three datasets, two scenarios (i.i.d. and OOD), and different buffer sizes. 'Mean' columns show average results for each dataset and across all datasets. The best results are in bold, second-best underlined. Note: The GDUMB method was excluded due to computational constraints.

| Task | | Split CIFAR-100 | | | | Split Mini-ImageNet | | | | Split Tiny-ImageNet | | | | Mean |
|---|---|---|---|---|---|---|---|---|---|---|---|---|---|---|
| Buffer | | 500 | 1000 | 2000 | Mean | 500 | 1000 | 2000 | Mean | 500 | 1000 | 2000 | Mean | |
| GEM | i.i.d. | -66.83 | -48.93 | -34.40 | -50.05 | -51.53 | -46.69 | -42.43 | -46.88 | -46.84 | -41.11 | -35.97 | -41.31 | -46.08 |
| | OOD | -18.85 | -15.75 | -12.81 | -15.80 | **-9.09** | **-8.33** | **-7.12** | **-8.18** | **-8.75** | -7.80 | **-6.59** | **-7.71** | **-10.57** |
| A-GEM | i.i.d. | -85.68 | -85.05 | -86.07 | -85.60 | -66.56 | -66.79 | -66.60 | -66.65 | -64.18 | -64.71 | -64.61 | -64.50 | -72.25 |
| | OOD | -22.42 | -22.16 | -22.39 | -22.32 | -12.56 | -12.54 | -12.46 | -12.52 | -12.66 | -12.81 | -12.75 | -12.74 | -15.86 |
| ER | i.i.d. | -81.09 | -77.19 | -71.21 | -76.50 | -67.16 | -65.12 | -61.56 | -64.61 | -70.61 | -69.68 | -67.10 | -69.13 | -70.08 |
| | OOD | -21.86 | -21.21 | -20.19 | -21.09 | -12.46 | -11.93 | -11.25 | -11.88 | -12.80 | -12.72 | -12.27 | -12.60 | -15.19 |
| GSS | i.i.d. | -70.26 | -61.81 | -53.51 | -61.86 | -61.72 | -55.95 | -46.80 | -54.82 | -67.56 | -64.05 | -58.06 | -63.22 | -59.97 |
| | OOD | **-16.85** | -15.86 | -13.79 | -15.50 | -11.43 | -10.57 | -8.80 | -10.27 | -11.64 | -11.00 | -9.74 | -10.79 | -12.19 |
| GDUMB | i.i.d. | - | - | - | – | - | - | - | – | - | - | - | – | – |
| | OOD | - | - | - | – | - | - | - | – | - | - | - | – | – |
| HAL | i.i.d. | -55.71 | -53.55 | -47.52 | -52.26 | **-38.88** | **-38.11** | **-36.63** | **-37.87** | **-34.95** | **-34.65** | **-32.17** | **-33.92** | -41.35 |
| | OOD | -17.53 | -15.95 | -14.51 | -16.00 | -10.85 | -10.09 | -9.38 | -10.11 | -10.40 | -9.86 | -8.88 | -9.71 | -11.93 |
| MetaSP | i.i.d. | -76.33 | -68.65 | -58.89 | -67.96 | -64.84 | -59.59 | -52.23 | -58.89 | -70.43 | -66.67 | -61.56 | -66.22 | -64.36 |
| | OOD | -20.37 | -18.52 | -16.89 | -18.59 | -11.34 | -10.19 | -9.05 | -10.19 | -11.41 | -11.61 | -10.68 | -11.58 | -13.45 |
| SOIF | i.i.d. | -71.47 | -62.91 | -62.11 | -65.50 | -65.70 | -58.65 | -54.76 | -59.70 | -69.63 | -65.65 | -63.64 | -66.31 | -63.84 |
| | OOD | -19.72 | -17.04 | -16.44 | -17.73 | -12.35 | -11.39 | -10.74 | -11.49 | -12.61 | -12.04 | -11.19 | -11.95 | -13.72 |
| Ours | i.i.d. | **-41.78** | **-35.08** | **-24.69** | **-33.85** | -49.71 | -41.82 | -37.03 | -42.85 | -49.27 | -44.69 | -36.96 | -43.64 | **-40.11** |
| | OOD | -18.38 | **-15.44** | **-7.85** | **-13.89** | -24.86 | -21.24 | -19.07 | -21.72 | -22.69 | -20.89 | -17.42 | -20.33 | -18.65 |

ER (18.75%), and GEM (18.68%)—all underperform in comparison. As shown in Table 2, our method also achieves an average BWT of -42.85%, outperforming MetaSP (-58.89%), GSS (-54.82%), SOIF (-59.70%), ER (-64.61%), and GEM (-46.88%). Thus, our method not only surpasses MetaSP, GSS, SOIF, ER, and GEM by margins of 4.19%, 5.51%, 6.67%, 9.61%, and 9.68% in terms of average ACC in the i.i.d. scenario but also outperforms them in average BWT by 16.04%, 11.97%, 16.85%, 21.76%, and 4.03%, respectively.

In the OOD scenario for both Split Mini-ImageNet and Split Tiny-ImageNet, our method doesn't surpass others. As illustrated in Table 1, the ACC for all methods is too low, indicating that they have not acquired sufficient knowledge to retain, let alone forget. This raises concerns about their OOD generalization capabilities.

For the i.i.d. scenario on Split Tiny-ImageNet, we observe a similar pattern as with Split Mini-ImageNet. The HAL method achieves the highest average BWT, followed by GEM. However, as shown in Table 1, HAL performs poorly, registering a significantly low average ACC of 2.65%. Similarly, GEM's ACC is also low at 6.69%. These results indicate that HAL and GEM struggle with knowledge retention, raising questions about their overall learning effectiveness. Thus, a fair comparison should again account for both BWT and ACC.

As detailed in Table 1, our method achieves an average ACC of 16.69% in the i.i.d. scenario on this dataset. This surpasses the next best methods, MetaSP (12.69%), GSS (12.07%), SOIF (10.78%), and ER (9.85%). Moreover, as indicated in Table 2, our method achieves an average BWT of -43.64%, outperforming MetaSP (-66.22%), GSS (-63.22%), SOIF (-66.31%), and ER (-69.13%). Therefore, our method not only surpasses these methods by margins of 4.0%, 4.62%, 5.91%, and 6.84% in average ACC, but also outperforms them in average BWT by 22.58%, 19.58%, 22.67%, and 25.49%, respectively, in the i.i.d. scenario on this dataset.

**Final class/task distribution in the buffer.** We analyze the buffer at the end of training for each method, extracting the number of samples per class and task. Subsequently, we calculate the Coefficient of Variation (CV) for classes and tasks using the formula $\left(\frac{\text{Standard Deviation}}{\text{Mean}}\right) \times 100\%$. The results are presented in Table 3. A CV of 0.00 signifies perfect balance, with higher values indicating greater imbalance. Our method demonstrates a perfectly balanced task distribution in the buffer with a CV of 0.00. Similarly, GEM and AGEM also achieve a CV of 0.00, indicating perfect task balance. ER, GDUMB and HAL achieve values close to zero, suggesting nearly balanced tasks. In contrast, GSS, MetaSP, and SOIF exhibit high CV values, indicating significant task imbalances.

Regarding class distribution, only one method (Ours) achieves a CV of 0.00, indicating perfect balance, while the remaining methods show high CV values, suggesting imbalanced class distribution within the buffer.

Table 3: Coefficient of Variation (CV) for class and task distributions in the buffer at the conclusion of training across different methods, using a buffer size of 1000 with Split CIFAR-100.

| Method | GEM | A-GEM | ER | GSS | GDUMB | HAL | MetaSP | SOIF | Ours |
|---|---|---|---|---|---|---|---|---|---|
| CV of Tasks | 0.00 | 0.00 | 06.80 | 58.79 | 05.10 | 06.21 | 24.49 | 29.87 | 0.00 |
| CV of Classes | 28.91 | 29.77 | 29.87 | 68.38 | 28.98 | 34.35 | 45.96 | 63.97 | 0.00 |

**Stability vs Plasticity.** In Figure 3, we analyze the performance of the compared methods across both i.i.d. and OOD scenarios at different training stages. Specifically, subfigures (b), (d), (f), and (h) show the average performance of seen tasks at each stage $t$, while subfigures (a), (c), (e), and (g) present the performance of each task at the end of training. In (a), depicting the i.i.d. scenario, existing methods demonstrate low accuracy on earlier tasks (indicating low stability) but high accuracy on later tasks (indicating high plasticity), resulting in relatively high average accuracy (ACC) across all tasks. In contrast, our method achieves higher accuracy on earlier tasks (greater stability) and balanced plasticity, resulting in a more balanced stability-plasticity trade-off. This pattern is also evident in subfigures (e), which covers the OOD scenario, where our method shows a larger margin of improvement. Notably, in (e), baseline methods perform poorly on all tasks except the last one, resulting in their ACC being driven solely by the final task.

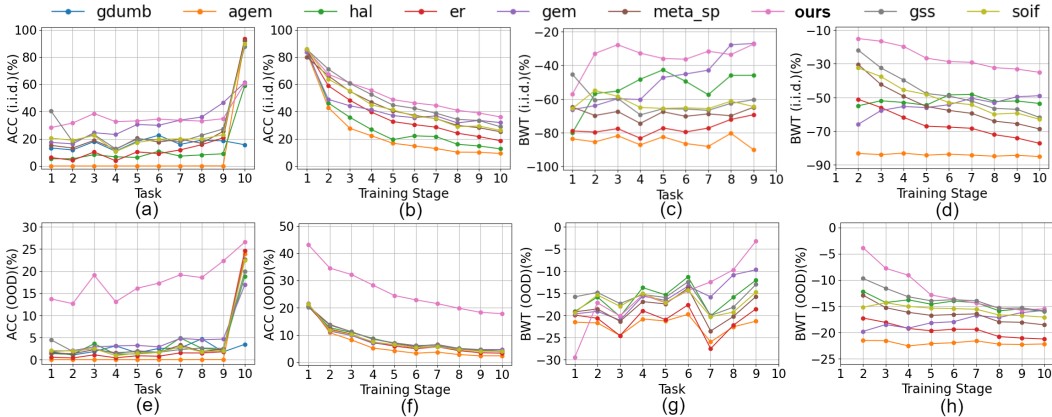

Figure 3: Analysis of various methods with a buffer size of 1000 on the Split CIFAR-100 dataset (results are averaged over 5 runs). This figure presents ACC and BWT for each method under both the i.i.d. and OOD scenarios at every training stage. Specifically, it shows the average performance of seen tasks at each stage $t$ and the performance of each task at the end of training.

In subfigure (b) (i.i.d. scenario), our method consistently outperforms others in ACC across training stages, except at stage 2, where the difference is minor. This trend is even more pronounced in subfigure (f) (OOD scenario), where our method shows a substantial margin of improvement. Subfigure (f) also illustrates that at each training stage (except the first), the OOD ACC of the baseline methods is very low. Consequently, as shown in subfiguret (g), the forgetting of the baseline methods in the OOD scenario is less severe than in the i.i.d. scenario (as shown in subfigure (c)), which leads to the baselines performing better in terms of forgetting in the OOD scenario. Therefore, for a fair comparison, as discussed earlier, both BWT and ACC should be considered together.

In subfigure (c), our method shows better BWT performance than the baselines in the i.i.d. scenario, except for tasks 1 and 8, where the difference is small. In subfigure (d), our method outperforms the other methods by a large margin in BWT at every training stage. As new tasks are added, retaining knowledge from earlier tasks becomes more challenging. However, as shown in subfigure (d), the rate of decline in performance for our method as tasks are added is smaller than that of the baselines, demonstrating better retention of previous task knowledge. In subfigure (h), our method shows slightly better BWT performance than the baselines in the OOD scenario. As discussed

earlier, since the baselines exhibit low ACC in the OOD scenario (as shown in subfigure (e) and (f)), they fail to retain or even learn much useful knowledge, leading to lower forgetting.

**Different Update Policies**   We analyze various update policies for our proxy-based contrastive learning method in Table 4. Initially, we evaluate the Reservoir update policy, which updates the buffer batch-by-batch using uniform random sampling. As seen in the ER approach (Table 3), which also employs Reservoir sampling, this policy can lead to class imbalances due to the batch-wise buffer update and random sampling. Despite the randomness of the Reservoir update, our method achieves higher ACC compared to the second-best method, GSS, by margins of 1.11% in the i.i.d. scenario and 8.16% in the OOD scenario, as shown in Table 1. This suggests that our proxy-based contrastive loss is effective in generating distinct representations.

Table 4: Performance comparison of different policy updates in our proxy-based contrastive learning method across both i.i.d. and OOD scenarios. The results represent the average of 5 runs using a buffer size of 500 on the Split CIFAR-100 dataset.

| Memory Update Policy | | Reservoir | Balanced Class/Task | | |
|---|---|---|---|---|---|
| | | | Random | Hard | Challenging |
| Accuracy | i.i.d. | 23.34 | 26.20 | 14.23 | 29.27 |
| | OOD | 11.61 | 12.79 | 7.20 | 14.81 |
| BWT | i.i.d. | -58.22 | -32.53 | -65.47 | -41.78 |
| | OOD | -34.38 | -13.27 | -33.04 | -18.38 |

However, the Reservoir update also results in higher forgetting, as observed in both the ER method (Table 2) and the Reservoir policy (Table 4). The batch-by-batch update increases the number of seen samples in the buffer, contributing to class imbalances. As the model encounters numerous samples, including outliers, it tends to focus less on learning informative features and more on memorizing samples due to the frequent buffer updates. This behavior increases the forgetting rate. To address this, we shift to a task-by-task buffer update strategy and aim for a balanced class/task distribution in the buffer. The results of this approach, labeled as "Balanced Class/Task," are shown in the table.

In the Random policy scenario, this change leads to significant improvements in both ACC and BWT, surpassing the Reservoir policy by a large margin. In particular, the BWT improves by 25.69% in the i.i.d. scenario and 21.11% in the OOD scenario. This demonstrates that transitioning from batch-by-batch to task-by-task updates and ensuring balanced class/task distributions reduce the forgetting rate.

Next, we analyze the effect of populating the buffer with misclassified samples. As previously mentioned, these samples fall into two categories: outliers (referred to as "Hard" in the table) and boundary samples (referred to as "Challenging" or "ACR"). Populating the buffer with Hard (low-confidence) samples yields the worst performance, whereas using Challenging (high-variability, boundary) samples improves ACC in both i.i.d. and OOD scenarios. Although the BWT in the Challenging scenario is slightly lower than in the Random policy, it still outperforms the Reservoir policy and shows better overall performance compared to other strategies in the table.

**Ablation.**   The ablation study, including our hyperparameter $E$ sensitivity analysis, running times, and resource usage (GPU and CPU) of various methods, is detailed in Appendix A.2.

## 5   CONCLUSION

Our work introduces novel contributions to the field of continual learning by addressing the challenges posed by distributional shifts. We provide the first demonstration that rehearsal-based methods significantly degrade when the i.i.d. assumption is violated, underscoring the need for approaches that can adapt to real-world scenarios. By incorporating contrastive learning and a dual optimization objective, our method optimizes both the encoder and classifier while ensuring that the buffer is populated with critical boundary samples, excluding outliers. Furthermore, our technique maintains a balanced class and task distribution in the buffer, leading to a more stable and robust learning process. Ultimately, we achieve a practical balance between stability and plasticity, while reducing resource consumption and computational time, making our approach both efficient and effective for real-world applications.

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

# A APPENDIX

## A.1 RELATED WORK

**Rehearsal-based CL Methods.** Rehearsal-based methods prevent catastrophic forgetting by maintaining a memory buffer of past data and periodically retraining the model with both old and new information. This approach enhances the model's ability to retain the knowledge of previous tasks Lopez-Paz & Ranzato (2017); Chaudhry et al. (2018b; 2019); Aljundi et al. (2019); Prabhu et al. (2020); Chaudhry et al. (2021); Sun et al. (2022; 2023).

GEM Lopez-Paz & Ranzato (2017) and its lightweight version A-GEM Chaudhry et al. (2018b) leverage previous training data to minimize gradient interference explicitly. They populate the buffer by randomly selecting samples, maintaining an equal number of samples for each task. ER Chaudhry et al. (2019) presents a straightforward rehearsal method that updates memories through reservoir sampling and employs random sampling during memory retrieval. Despite its simplicity, it remains a strong baseline. GSS Aljundi et al. (2019) approaches updating the memory buffer as a constrained optimization problem, to maximize the diversity of sample gradients within the buffer. GDUMB Prabhu et al. (2020) operates by greedily accumulating new training samples into the buffer, which it subsequently uses to train a model from scratch for testing. HAL Chaudhry et al. (2021) employs the old training samples, which are more susceptible to forgetting, as "anchors" to stabilize their predictions. This method enhances replay by adding an objective that minimizes the forgetting of crucial learned data points. MetaSP Sun et al. (2022) utilizes the Pareto optimum of example influence on stability and plasticity, thereby guiding updates to the model and storage management. SOIF Sun et al. (2023) leverages second-order influences to make more informed decisions about which samples to retain in the buffer.

**Data-centric.** Data-centric approaches emphasize the quality of data over the complexity of models. Data-centric artificial intelligence involves techniques aimed at enhancing datasets, thereby enabling the training of models with fewer data requirements Motamedi et al. (2021); Mazumder et al. (2022). Ignoring the critical importance of data has led to inaccuracies, biases, and fairness issues in real-world applications Mazumder et al. (2022). High-quality data can significantly improve model generalization, mitigate bias, and enhance safety in data cascades Sambasivan et al. (2021); Aroyo et al. (2022). For instance, Toneva et al. (2018) and Swayamdipta et al. (2020) leverage model confidence during training to cleanse the dataset. Specifically, Swayamdipta et al. (2020) categorizes the dataset into easy-to-learn, hard-to-learn, and ambiguous subsets, whereas Toneva et al. (2018) distinguishes between forgettable and unforgettable samples. These methods allow for the assessment of each sample's contribution during training. Building on these approaches, we introduce ACR to identify boundary samples and populate the buffer with them.

**Proxy-based Contrastive Learning.** A frequently employed approach is softmax cross-entropy loss, where proxies stand in for classes. This approach calculates the similarity between each anchor and the proxies across $C$ classes Chaudhry et al. (2019); Sun et al. (2023). Proxies represent subdatasets, while the anchor is one of the samples within the training batch Yao et al. (2022); Lin et al. (2023). Supervised contrastive loss, in contrast, builds positive pairs from samples within the same class, evaluating the similarity between each anchor and all $N$ samples in the training batch. This type of loss is usually combined with the nearest class mean classifier Mensink et al. (2013); Mai et al. (2021).

Contrastive-based loss primarily explores detailed sample-to-sample relationships, while proxy-based loss relies on proxies to represent subsets of the training data, leading to faster and more stable convergence but potentially overlooking some semantic relationships. To address these limitations, Yao et al. (2022) and Lin et al. (2023) introduced proxy-based losses that incorporate key advantages of contrastive learning, improving domain generalization and enabling online class-incremental learning, where each task is trained for only one epoch.

## A.2 ABLATION

**Hyperparameter sensitivity.** In Figure 4, we examine the sensitivity of our hyperparameter $E$, which is integral to our memory update policy. We use the reservoir update method as the baseline for comparison. Both update policies employ our proxy-based contrastive learning approach. As illustrated, the performance of hyperparameter $E$ remains stable across a range from 2 to 7 on all three datasets, maintaining consistency in both ACC and BWT under i.i.d. and OOD conditions. Additionally, the results indicate that our approach for identifying boundary samples requires only a few initial training epochs.

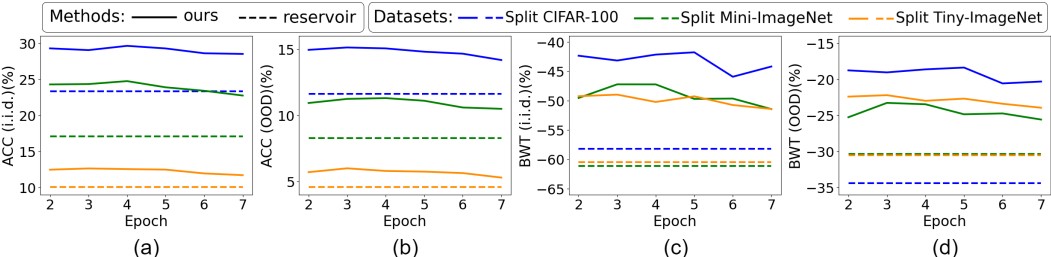

Figure 4: Hyperparameter sensitivity analysis of $E$ in our boundary sample identification and buffer population approach. The evaluation is performed across all three datasets with a buffer size of 500, using both ACC and BWT metrics, under i.i.d. and OOD scenarios. Results are averaged over 5 runs. Each color represents a dataset: solid lines indicate our updating policy, while dashed lines correspond to the reservoir update policy. Both policies utilize our proxy-based contrastive learning method, with only the update policy varying.

**Running Time.** Table 5 presents a comparison of the running time of each method per hour. Among the high-ACC methods (GEM, GSS, MetaSP, SOIF), our method is slower only than MetaSP. However, when considering GPU memory usage (see Figure 5), our method requires approximately 1.3 GB, whereas MetaSP consumes nearly 11.8 GB—about 9 times more. This significant resource demand limits MetaSP's applicability in environments with constrained resources. In contrast, GEM, GSS, and SOIF have much longer running times. While A-GEM, ER, and HAL are faster, their performance suffers from significantly lower ACC.

Table 5: Running times per hour for different methods on Split CIFAR-100 (buffer size 2000) using Quadro RTX 8000.

| Method | GEM | A-GEM | ER | GSS | HAL | MetaSP | SOIF | Ours |
|---|---|---|---|---|---|---|---|---|
| **Runtime (hours)** | 20.42 | 1.85 | 1.25 | 15.56 | 2.44 | 2.8 | 8.13 | 3.49 |

**Memory usage.** Figure 5 illustrates the resource utilization for each method up to the 35k time step, with specific values shown at the 5k time step for comparison. For GPU memory, our method consumes approximately 1.3 GB, similar to lower-ACC methods such as ER, AGEM, and HAL. In contrast, higher-ACC methods like MetaSP, SOIF, GSS, and GEM use significantly more memory—about 11.8 GB, 10.2 GB, 10.2 GB, and 10.2 GB, respectively. For CPU usage, our method requires around 18%, aligning with the lower-ACC methods, whereas MetaSP uses about 32%, and SOIF, GSS, and GEM each use around 80%. These data signify that our method is not only more resource-efficient, similar to low-ACC methods, but also surpasses the performance of high-ACC methods.

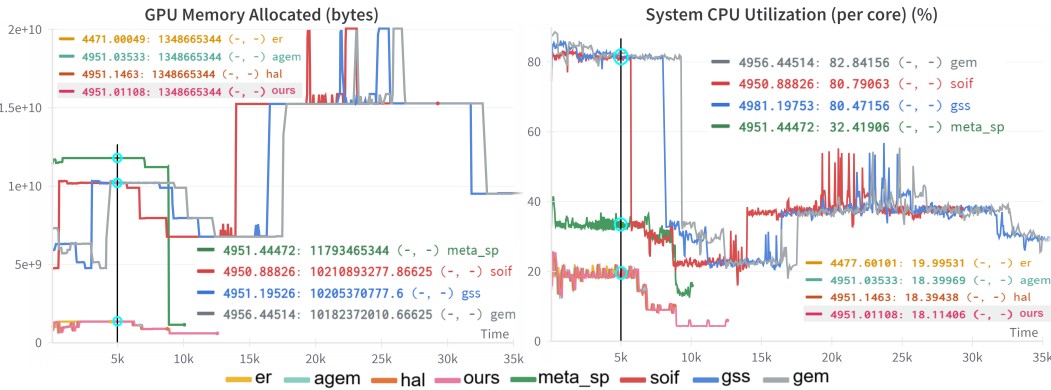

Figure 5: GPU usage and CPU utilization for each method up to 35k time steps, with detailed values at 5k steps. The experiment was conducted with Split CIFAR-100, a buffer size of 2000.

