# OpenReview forum: "Continual Learning: Less Forgetting, More OOD Generalization via Adaptive Contrastive Replay"
_ICLR.cc/2025/Conference — Submitted to ICLR 2025_

### Official Review · Reviewer_BhyB · 2024-10-27

**Soundness:** 3
**Presentation:** 3
**Contribution:** 3
**Rating:** 6
**Confidence:** 3

**Summary:**

This paper introduces Adaptive Contrastive Replay (ACR), a novel rehearsal-based continual learning method aimed at improving Out-of-Distribution (OOD) generalization while mitigating catastrophic forgetting. ACR leverages proxy-based contrastive learning and confidence-guided sample selection to populate its buffer with boundary samples rather than outliers. This approach balances stability and plasticity, ensuring the retention of old knowledge while adapting to new information. ACR achieves superior performance across multiple benchmarks (Split CIFAR-100, Split Mini-ImageNet, and Split Tiny-ImageNet) in both i.i.d. and OOD settings, outperforming state-of-the-art rehearsal-based methods.

**Strengths:**

1. Relevant Contribution: Tackles the overlooked issue of OOD generalization in continual learning, which enhances the practical applicability of CL methods.
2. Balanced Stability and Plasticity: Focuses on boundary sample selection, which helps maintain a stable yet adaptive learning process.
3. Efficiency Gains: Reduces computational complexity, making ACR suitable for deployment in resource-constrained environments.
4. Comprehensive Evaluation: Provides comparisons across multiple datasets and with several rehearsal-based baselines, showcasing consistent improvements.

**Weaknesses:**

1. Narrow Scope of Baselines: The paper compares ACR only with other rehearsal-based approaches, omitting non-rehearsal continual learning methods that could offer additional insights.
2. Over-reliance on Specific OOD Corruptions: The focus on synthetic corruptions (e.g., Gaussian noise, blur) may not fully represent real-world OOD scenarios such as domain shifts or adversarial inputs.
3. Limited Hyperparameter Analysis: The impact of key hyperparameters like the temperature $\tau$ and buffer size is not thoroughly explored, leaving some questions about robustness unanswered.
4.Scaling Limitations: There is insufficient discussion about how ACR performs on longer task sequences or more complex datasets beyond the three benchmarks used in the paper.
5. Clarity Issues: Figure 2 is unclear, and key elements (e.g., proxy-based contrastive loss) would benefit from more intuitive explanations.
6.Assumptions in Methodology: The paper assumes that high-variance boundary samples will always improve performance, without discussing scenarios where this assumption might fail (e.g., noisy labels or unbalanced datasets).

**Questions:**

1. Figure 2 is unclear. Can the authors provide an improved version or additional explanations to clarify how the buffer update mechanism works?
How would ACR perform under more realistic OOD scenarios such as domain shifts, adversarial attacks, or real-world data variability?
2. What are the computational trade-offs when balancing between boundary and outlier samples? Can ACR scale effectively for larger datasets or continuous tasks?
3. How sensitive is the performance to the temperature parameter ($\tau$) in the proxy-based contrastive loss, and what is the impact of different buffer sizes on the results?
4. Are there any failure cases or situations where ACR struggles, particularly for longer task sequences or when facing noisy or imbalanced datasets?

---

> ### Author Response · Authors · 2024-11-28
>
> Thank you for your insightful comment We appreciate the opportunity to clarify your concerns.
>
> **Answer to W1**
>
> Buffer-based methods store samples in a buffer, which has been shown in the literature to yield better performance compared to other continual learning (CL) approaches. However, this advantage comes with the drawback of memorizing samples, making these methods more prone to overfitting, as noted in prior work. For this reason, our focus has been primarily on buffer-based methods. Nonetheless, as per your suggestion, we will include results for architecture- and regularization-based methods in the camera-ready version.
>
>
> **Answer to W2**
>
> To address the challenge of over-reliance on synthetic OOD corruptions and better emulate real-world distribution shifts, we draw on the methodology proposed by Hendrycks and Dietterich [1]. Their work introduces a standardized benchmark designed to evaluate the robustness of image classifiers against a variety of common corruptions and perturbations, offering a more comprehensive and systematic approach to studying robustness. This benchmark not only identifies classifiers that are better suited for safety-critical applications but also broadens the focus of robustness research beyond adversarial examples. By emphasizing common real-world corruptions, the framework bridges the gap between synthetic benchmarks and practical robustness requirements, enabling a more realistic evaluation of classifier performance under OOD conditions.
>
> To further extend this work, we propose addressing domain shifts and adversarial inputs as part of future research directions.
>
> [1] Dan Hendrycks and Thomas Dietterich. Benchmarking neural network robustness to common corruptions and perturbations. arXiv preprint arXiv:1903.12261, 2019.
>
>
> **Answer to W3**
>
> For the temperature \(\tau\) hyperparameter, we follow prior studies and select a small value of 0.09, as commonly done in the literature. The role of the temperature \(\tau\) in contrastive loss has been extensively analyzed in previous works. However, if further analysis is deemed necessary, we are prepared to conduct a sensitivity study within our framework and include the results in the camera-ready version.
>
> Regarding buffer sizes, we conducted experiments with various configurations, including 500, 1000, and 2000, to evaluate the impact on performance.

---

> ### Author Response · Authors · 2024-11-29
>
> **Answer to W4**
>
> Regarding task sequences, we evaluate our approach across several stages:
>
> 1. **Split CIFAR-100**: This dataset consists of 100 classes divided into 10 tasks, each containing 10 classes.
> 2. **Split Mini-ImageNet**: This dataset divides 100 classes into 5 tasks, each with 20 classes.
> 3. **Split Tiny-ImageNet**: This dataset includes 200 classes split into 10 tasks, each containing 20 classes.
>
> Each dataset provides 500 training samples per class. The number of test samples varies: Split CIFAR-100 and Split Mini-ImageNet have 100 test samples per class, while Split Tiny-ImageNet has 50 test samples per class.
>
> Regarding dataset complexity, **Split Mini-ImageNet** and **Split Tiny-ImageNet** are recognized as challenging benchmarks in the continual learning literature. While some studies use simpler datasets like MNIST or CIFAR-10, Split Mini-ImageNet, and Split Tiny-ImageNet are considered more complex and are often used to benchmark methods under more demanding conditions.
>
>
> **Answer to W5**
>
> We will enhance Figure 2 by incorporating your suggestion and ensuring it is updated for the camera-ready version.

---

> ### Author Response · Authors · 2024-11-29
>
> **Answer to W6**
>
>
> Thank you for highlighting this important question about the robustness of confidence variance.
>
>
> To identify the most informative samples of each class, we utilized a data-centric artificial intelligence approach. In data-centric AI, the emphasis is on improving datasets to facilitate effective model training with fewer data requirements [1, 2]. For instance, as cited in our paper, [3] and [4] leverage model predictions during training to refine datasets. These studies demonstrate that early-stage uncertainty is a reliable indicator of a sample’s contribution to the training process. By analyzing network predictions for individual samples during the initial training epochs, they identify noisy labels and outlier samples, aiding dataset optimization.
>
>
> During the training process, the decision boundary evolves. The key question is: **Which boundary leads to better generalization?** One shaped by outliers or one influenced by genuinely challenging (most informative) samples?
>
>
> To address this, we analyzed CIFAR10 in batch training (training all 10 classes at once, resembling one task of Split CIFAR-100 under continual learning). We calculated the target confidence of each sample at every epoch and displayed the mean and variance of sample confidence across these epochs in the Figure linked below. In the left sub-figure, we consider the first \\(E\\) epochs \\(E = 5\\) as in our paper, and in the right sub-figure, we use 50 epochs (matching the epoch count per task in our paper).
>
> [Please click to view the Figure](https://anonymous.4open.science/r/Figures-EB2E/1.png)
>
> We categorized the samples observed during the first \\(E = 5\\) epochs into three groups:
>
>    - **Area 1.** **Simple Samples:** High confidence with low variance, consistently well-classified and at the center of the distribution.
>
>    - **Area 2.** **Challenging Samples (Most Informative):** Moderate mean confidence with high variance, residing near the decision boundary. They are ideal for shaping a robust boundary.
>
>    - **Area 3.** **Hard (Outlier) Samples:** Low confidence with low variance, far from the decision boundary, offering limited value for boundary refinement.
>
> The bottom sub-figure shows representative images from these areas (Area 1, Area 2, and Area 3) for each CIFAR10 class. Each column represents a class, with rows corresponding to simple, challenging, and hard samples, respectively. Simple samples display clear class attributes (e.g., a horse on grass; grappling with the effects of spurious correlation). Challenging samples exhibit ambiguous traits (e.g., an airplane against a gray backdrop; enhancing OOD generalization), while hard samples feature unusual or rare characteristics (e.g., a horse’s head only), often functioning as outliers.
>
> As training progresses from \\(E = 5\\) to \\(E = 50\\):
>
>    - **Simple Samples** retain high confidence and low variance, remaining at the center of the distribution.
>
>    - **Challenging Samples** show increasing confidence (mean confidence rises) and decreasing variance as the model learns them robustly. Most migrate from the boundary to the center, while a few persist near the boundary.
>
>    - **Hard Samples** exhibit higher confidence but also increased variance, eventually achieving moderate confidence and defining the boundary.
>
> Thus:
>
>    - **Area 4** (\\(E = 50\\)) contains simple samples and most challenging samples.
>
>    - **Area 5** (\\(E = 50\\)) contains hard samples and some challenging samples, shaping the boundary.
>
> Comparing Area 2 (\\(E = 5\\)) and Area 5 (\\(E = 50\\)), we argue that Area 2, with challenging samples, provides better generalization. By prioritizing these **challenging (most informative) samples** for buffer selection, the decision boundary is shaped by meaningful boundary samples rather than outlier samples. This strategy "reminds" the model of an ideal boundary in future tasks, promoting generalization and stability-plasticity balance.
>
> In Figure 4 of the Appendix, we analyze the sensitivity of the hyperparameter \\(E\\). The results show stable performance across a range (\\(E = 2\\) to \\(7\\)) on all datasets, maintaining consistency in ACC and BWT under i.i.d. and OOD conditions. These findings confirm that identifying boundary samples requires only a few initial training epochs.
>
>
> Additionally, as shown in the Figure linked below, we conducted another experiment on CIFAR-100, extending the range of \\(E\\) from 2 to 14. The results demonstrate that performance gradually decreases beyond \\(E = 7\\). This is because outliers transition to challenging samples, and challenging samples to simple ones, reducing their impact on refining the decision boundary.
>
>
>
> [Please click to view the Figure](https://anonymous.4open.science/r/Figures-EB2E/2.png)

---

> ### Author Response · Authors · 2024-11-29
>
> [1] Mohammad Motamedi, Nikolay Sakharnykh, and Tim Kaldewey. A data-centric approach for training deep neural networks with less data. arXiv preprint arXiv:2110.03613, 2021.
>
> [2] Mark Mazumder, Colby Banbury, Xiaozhe Yao, Bojan Karlas, William Gaviria Rojas, Sudnya Diamos, Greg Diamos, Lynn He, Alicia Parrish, Hannah Rose Kirk, et al. Dataperf: Benchmarks for data-centric ai development. arXiv preprint arXiv:2207.10062, 2022.
>
> [3] Mariya Toneva, Alessandro Sordoni, Remi Tachet des Combes, Adam Trischler, Yoshua Bengio, and Geoffrey J Gordon. An empirical study of example forgetting during deep neural network learning. arXiv preprint arXiv:1812.05159, 2018.
>
> [4] Swabha Swayamdipta, Roy Schwartz, Nicholas Lourie, Yizhong Wang, Hannaneh Hajishirzi, Noah A Smith, and Yejin Choi. Dataset cartography: Mapping and diagnosing datasets with training dynamics. arXiv preprint arXiv:2009.10795, 2020.

---

> ### Author Response · Authors · 2024-11-29
> **imbalanced datasets**
>
> Thank you once again for your thoughtful comments and the score you provided. we think we have addressed all your concerns in our previous responses. If we have overlooked anything or if you have additional feedback, please feel free to let us know so we can address them in the camera-ready version. Regarding imbalanced datasets, we will conduct the necessary experiments and include the results in the camera-ready version.

---

### Official Review · Reviewer_M5Un · 2024-11-01

**Soundness:** 3
**Presentation:** 3
**Contribution:** 2
**Rating:** 5
**Confidence:** 5

**Summary:**

The paper proposes a novel approach called Adaptive Contrastive Replay (ACR) to improve continual learning, specifically focusing on reducing forgetting and enhancing out-of-distribution (OOD) generalization. Existing methods for continual learning often rely on retaining past samples to prevent catastrophic forgetting but struggle with OOD generalization, leading to imbalances and high memory costs. ACR addresses this by using a dual optimization strategy that refines the decision boundary, focusing on informative boundary samples rather than simple memorization. By balancing class and task representation in the memory buffer and leveraging a proxy-based contrastive loss, ACR achieves significantly better stability-plasticity trade-offs and outperforms prior methods on various benchmarks like Split CIFAR-100 and Mini-ImageNet

**Strengths:**

1. OOD generalization: it combines the robust features of contrastive learning with adaptive memory buffer management to address poor OOD generalization in continual learning.
2. Enhanced Memory Management: The method’s strategy for memory buffer management, which focuses on maintaining a balanced distribution of classes and tasks, helps mitigate the long-tail effect often observed in other methods. This leads to more stable learning outcomes.

**Weaknesses:**

1. Lack of novelty: The proposed "Proxy-Based Contrastive Loss" is already widely used in Continual Learning, as shown in [1], [2]. Please clarify the unique aspects of your method in comparison to existing approaches.
2. Lack of ablation study: In your paper, you propose using the output of the last layer as an index to estimate image uncertainty, which then guides the management of your memory set. I believe this could help select images near the decision boundary. Based on my experience, combining data augmentation and contrastive learning generally results in significant accuracy improvements in continual learning. Could you specify the performance gain achieved by solely using your memory management method? Additionally, what are the results when other methods are combined with your contrastive learning loss?


[1] De Lange, Matthias and Tuytelaars, Tinne. "Continual Prototype Evolution: Learning Online From Non-Stationary Data Streams", ICCV,2021.
[2] Lucas Caccia, Rahaf Aljundi, Nader Asadi, Tinne Tuytelaars, Joelle Pineau, Eugene Belilovsky. "New Insights on Reducing Abrupt Representation Change in Online Continual Learning", ICLR,2022.

**Questions:**

See weaknesses.

---

> ### Author Response · Authors · 2024-11-25
> **Answer to Q1 (Comparison with CoPE [1])**
>
> Thank you for your insightful comment. We appreciate the opportunity to clarify the differences between our approach and the CoPE [1], as well as ER-AML, and ER-ACE [2]. Our proposed method, **Adaptive Contrastive Replay (ACR)**, introduces significant novelties that differentiate it from existing approaches.
>
> Below, we provide a detailed comparison between our approach and CoPE [1]:
>
>
> ### **Proxy in ACR (Adaptive Contrastive Replay):**
>
> 1. **Definition:**
>    - In ACR, the **weights of the classification layer (linear layer)** are used as proxies for each class. Each class \\( y \\) is represented by a weight vector \\( W_y \\in \\mathbb{R}^d \\), which serves as a class-level anchor in the latent space.
>
> 2. **Formulation:**
>    - The proxy-based contrastive loss aligns the latent embeddings \\( f_\theta(x) \\) of inputs with their respective class weights \\( W_y \\) while pushing them away from the weights of other classes:
>      \\[
>      L_{\text{PCL}} = - \frac{1}{N} \sum_{i=1}^N \log \frac{\exp(f_\theta(x_i)^\top W_{y_i} / \tau)}{\sum_{L \in \mathcal{C}} \exp(f_\theta(x_i)^\top W_L / \tau)}.
>      \\]
>      - \\( W_{y_i} \\): Weight vector (proxy) for the target class \\( y_i \\).
>      - \\( \\mathcal{C} \\): Set of class labels in the batch.
>      - \\( \\tau \\): Temperature parameter.
>
> 3. **Role of Weights:**
>    - The weights (\\( W_y \\)) act as **class-level representatives** in the latent space.
>    - These proxies guide the embeddings \\( f_\\theta(x) \\) toward their respective classes, ensuring better class separation in the latent space.
>
> 4. **Classifier:**
>    - **Linear Cosine Similarity Classifier:** For ACR, the classification is performed using a linear cosine similarity-based classifier. This computes the cosine similarity between normalized feature embeddings and normalized class weights (proxies), scaled by a temperature parameter:
>
>      \\[
>      \text{logits}[y] = \frac{\cos\left(f_\theta(x), W_y\right)}{\tau},
>      \\]
>
>      where:
>      -  \\( f_\\theta(x) \\) is the normalized feature embedding of the input \\( x \\),
>      -  \\( W_y \\) is the normalized weight vector (proxy) for class \\( y \\),
>      -  and \\( \tau \\) is the temperature parameter controlling the sharpness of the logits.
>
>      The class with the highest logit is chosen as the predicted class:
>
>      \\[
>      \hat{y} = \arg\max_{y \in \mathcal{Y}} \text{logits}[y].
>      \\]
>
> 5. **Dynamic Behavior:**
>    - The proxies (\\( W_y \\)) are tied to the classifier's weight matrix and evolve automatically during training without requiring additional memory structures.
>
> ---
>
> ### **Proxy in CoPE (Continual Prototype Evolution):**
>
> 1. **Definition:**
>    - CoPE maintains **explicit prototypes** for each class \\( c \\), denoted as \\( p_c \\in \\mathbb{R}^d \\). These prototypes are stored in a separate memory and represent the **center of mass** of the latent embeddings for their respective classes.
>
> 2. **Formulation:**
>    - Prototypes are updated dynamically using a momentum-based rule:
>      \\[
>      p_c \gets \alpha p_c + (1 - \alpha) \bar{p}_c,
>      \\]
>      - \\( \\alpha \\): Momentum parameter.
>      - \\( \\bar{p}\_c = \\frac{1}{|B_c|} \\sum_{x_i \\in B_c} f_\\theta(x_i) \\): Mean of latent embeddings for class \\( c \\) in the current batch \\( B_c \\).
>      - After updating, the prototype \\( p_c \\) is normalized to unit length:
>      \\[
>      p_c \gets \frac{p_c}{\||p_c\||}.
>      \\]
>
>
> 3. **Role of Prototypes:**
>    - The prototypes explicitly represent the **cluster center** for each class in the latent space.
>    - They are independent of the classifier and stored in a separate memory structure.
>    - Prototypes are used in the pseudo-prototypical proxy loss (PPP-loss) to enforce intra-class compactness and inter-class separation.
>
> 4. **Classifier:**
>    - **Nearest Neighbor Classifier:** CoPE uses a nearest neighbor classifier for classification. It compares the feature embedding \\( f_\\theta(x) \\) to stored prototypes \\( p_c \\) and predicts the class based on the closest prototype:
>      \\[
>      c^* = \arg\max_{c \in Y} \cos(f_\theta(x), p_c)
>      \\]
>
> 5. **Dynamic Behavior:**
>    - The prototypes are explicitly updated with each batch of data to reflect changes in class distributions over time.
>
> ---

---

> > ### Author Response · Authors · 2024-11-25
> > **Key Differences between Ours and CoPE**
> >
> > ### **Key Differences**:
> >
> > | Method                | **ACR**                                             | **CoPE**                                             |
> > |------------------------|----------------------------------------------------|----------------------------------------------------|
> > | **Proxy Type**          | Weights of the classification layer.               | Explicit prototypes stored in memory.              |
> > | **Purpose**             | Represent class embeddings for alignment.          | Represent the center of mass of class distributions in latent space. |
> > | **Update Mechanism**    | Implicitly updated during classifier training.      | Explicitly updated using momentum-based averaging. |
> > | **Storage**             | No separate memory; tied to classifier weights.     | Stored explicitly in a prototype memory.           |
> > | **Classifier**          | Linear Cosine Similarity Classifier.   | Nearest neighbor classifier using stored prototypes. |
> > | **Usage in Loss**       | Proxy-based contrastive loss aligns embeddings with class weights. | PPP-loss aligns embeddings with prototypes while separating classes. |
> > | **Dynamic Nature**      | Evolves automatically as the classifier is trained.| Continuously updated to reflect new class distributions. |

---

> ### Author Response · Authors · 2024-11-25
> **Answer to Q1 (Comparison with ER-AML, and ER-ACE [2])**
>
> Below, we present a detailed comparison of our approach with ER-AML and ER-ACE [2]:
>
> ### **ACR Loss: Proxy-Based Contrastive Loss**
> - **Formulation**: Aligns the latent embedding \\( f_\theta(x) \\) (where \\( f_\\theta \\) is the encoder with parameters \\(\\theta\\), mapping the input \\(x\\) into the latent space) of a sample \\(x\\) with its class proxy \\(W_y\\) (a representative vector for the target class \\(y\\)) while separating it from proxies \\(W_L\\) of all other classes in the batch.
>   \\[
>   L_{\text{PCL}} = - \frac{1}{N} \sum_{i=1}^N \log \frac{\exp(f_\theta(x) \cdot W_y / \tau)}{\sum_{L \in C} \exp(f_\theta(x) \cdot W_L / \tau)}
>   \\]
>   - **\\(N\\)**: The total number of samples in the batch.
>   - **\\(\\tau\\)**: A temperature parameter controlling the sharpness of similarity distributions.
>   - **\\(C\\)**: The set of all class labels in the current batch.
>
> - **Objective**: Ensures compact clustering of embeddings for the same class and separation between clusters of different classes.
> - **Mechanism**:
>   - Optimizes the proxy-to-sample relationship for efficiency, reducing computational overhead by avoiding explicit pairwise comparisons between individual samples.
>   - Expands the pool of negatives by considering all non-target class proxies \\(W_L\\) as negatives.
> - **Key Focus**: Efficient and stable feature alignment using proxies.
>
> ---
>
> ### **ER-AML Loss: Asymmetric Metric Learning Loss**
> - **Formulation**: Combines two components:
>   1. **Supervised Contrastive Loss (SupCon)** for incoming data (\\(X_\text{in}\\), the set of new samples being learned):
>      \\[
>      L_1(X_\text{in}) = - \sum_{x_i \in X_\text{in}} \frac{1}{|P(x_i)|} \sum_{x_p \in P(x_i)} \log \frac{\text{sim}(f_\theta(x_p), f_\theta(x_i))}{\sum_{x_n \in N \cup P(x_i)} \text{sim}(f_\theta(x_n), f_\theta(x_i))}
>      \\]
>      - **\\(P(x_i)\\)**: The set of positive samples (other samples of the same class as \\(x_i\\) in the batch or buffer).
>      - **\\(N\\)**: The set of negative samples (samples from other classes in the batch or buffer).
>      - **\\(\\text{sim}(a, b)\\)**: The similarity between embeddings \\(a\\) and \\(b\\), calculated as:
>        \\[
>        \text{sim}(a, b) = \exp\left(\frac{a^\top b}{\tau \||a\|| \||b\||}\right)
>        \\]
>   2. **Cross-Entropy Loss** for replayed data (\\(X_\\text{bf}\\), the set of samples replayed from the buffer):
>      \\[
>      L_2(X_\text{bf}) = - \sum_{x \in X_\text{bf}} \log \frac{\text{sim}(W_{c(x)}, f_\theta(x))}{\sum_{c \in C_\text{all}} \text{sim}(W_c, f_\theta(x))}
>      \\]
>      - **\\(W_{c(x)}\\)**: The class proxy of the label \\(c(x)\\) corresponding to sample \\(x\\).
>      - **\\(C_\\text{all}\\)**: The set of all class labels observed so far.
>
> - **Combined Loss**:
>   \\[
>   L(X_\text{in} \cup X_\text{bf}) = \gamma L_1(X_\text{in}) + L_2(X_\text{bf})
>   \\]
>   - **\\(\\gamma\\)**: A weighting factor balancing the two terms.
>
> - **Objective**: Aligns embeddings of current task data while consolidating knowledge across previous tasks.
> - **Mechanism**:
>   - Uses pairwise relationships among samples (via \\(P(x_i)\\) and \\(N\\)) for finer control over embeddings.
>   - Incorporates class prototypes for replayed data to maintain stability.
>
> ---
>
> ### **ER-ACE Loss: Asymmetric Cross-Entropy Loss**
> - **Formulation**: Combines two cross-entropy losses:
>   \\[
>   L_{\text{ER-ACE}} = L_{\text{ce}}(X_\text{in}, C_\text{curr}) + L_{\text{ce}}(X_\text{bf}, C_\text{old} \cup C_\text{curr})
>   \\]
>   - **\\(C_\\text{curr}\\)**: The set of classes in the incoming batch.
>   - **\\(C_\\text{old}\\)**: The set of previously seen classes stored in the buffer.
>   - **\\(L_{\\text{ce}}(X, C)\\)**: The cross-entropy loss for samples \\(X\\) over classes \\(C\\):
>     \\[
>     L_{\text{ce}}(X, C) = - \sum_{x \in X} \log \frac{\exp(\text{sim}(W_y, f_\theta(x)))}{\sum_{c \in C} \exp(\text{sim}(W_c, f_\theta(x)))}
>     \\]
>
> - **Mechanism**:
>   - For \\(X_\\text{in}\\): Computes logits for only current classes \\(C_\\text{curr}\\), isolating new class learning.
>   - For \\(X_\\text{bf}\\): Includes both current and past classes (\\(C_\\text{curr} \\cup C_\\text{old}\\)), consolidating previous knowledge.
>
> - **Objective**: Indirectly enforces contrast by isolating logits for incoming data and leveraging replayed data for broader class alignment.
>
> ---

---

> > ### Author Response · Authors · 2024-11-25
> > **Key Differences Across Losses (Ours, ER-AML, and ER-ACE)**
> >
> > ### **Key Differences Across Losses**
> > 1. **Contrastive Mechanism**:
> >    - ACR explicitly aligns embeddings with proxies (\\(W_y\\)), optimizing proxy-based relationships.
> >    - ER-AML uses pairwise sample relationships, leveraging \\(P(x_i)\\) and \\(N\\).
> >    - ER-ACE induces contrast implicitly via the asymmetric treatment of logits.
> >
> > 2. **Negative Sample Selection**:
> >    - ACR: Considers all non-target proxies as negatives, broadening coverage.
> >    - ER-AML: Relies on sample-based negatives within the batch or buffer.
> >    - ER-ACE: Limits logits for negatives to the class sets \\(C_\\text{curr}\\) and \\(C_\\text{old}\\).
> >
> > 3. **Optimization Complexity**:
> >    - ACR: Lower due to proxy-based design.
> >    - ER-AML: Higher due to pairwise sample-to-sample comparisons.
> >    - ER-ACE: Moderate, as it focuses on logits rather than explicit pairwise comparisons.

---

> ### Author Response · Authors · 2024-11-25
> **Answer to Q2 (Memory management gain)**
>
> We appreciate the reviewer’s feedback and agree that additional analysis can clarify the contributions of each component of our method. To isolate the performance gain from our memory management strategy, we analyzed various update policies for our proxy-based contrastive learning method on the Split CIFAR-100 dataset with buffer sizes of 500, 1000, and 2000, as well as their mean performance. The results for each buffer size and the mean are presented in the following table.
>
> Initially, we evaluated the Reservoir update policy, which updates the buffer batch-by-batch using uniform random sampling. As observed in the ER approach (Table 3), which also employs Reservoir sampling, this policy can result in class imbalances due to batch-wise updates and random sampling. Despite the randomness of the Reservoir update, our method achieves a higher average ACC compared to the second-best method, GEM (i.i.d.: 29.86%; OOD: 4.34%; Table 1), by 1.47% in the i.i.d. scenario and 10.2% in the OOD scenario. This indicates that our proxy-based contrastive loss effectively generates distinct representations.
>
> However, the Reservoir update also results in higher forgetting, as observed in both the ER method (Table 2) and the Reservoir policy (in the following table). The batch-by-batch update increases the number of seen samples in the buffer, contributing to class imbalances. As the model encounters numerous samples, including outliers, it tends to focus less on learning informative features and more on memorizing samples due to the frequent buffer updates. This behavior increases the forgetting rate. To address this, we shifted to a task-by-task buffer update strategy and aimed for a balanced class/task distribution in the buffer. The results of this approach, labeled as ”Balanced Class/Task,” are shown in the following table.
>
> In the Random policy scenario, this change leads to significant improvements in both ACC and BWT, surpassing the Reservoir policy by a large margin. This demonstrates that transitioning from batch-by-batch to task-by-task updates and ensuring balanced class/task distributions reduce the forgetting rate and enhance accuracy.
>
> Next, we analyzed the effect of populating the buffer with misclassified samples. As previously mentioned, these samples fall into two categories: outliers (referred to as ”Hard” in the table) and boundary samples (referred to as ”Challenging” or ”ACR”). Populating the buffer with Hard (low-confidence) samples yields the worst performance, whereas using Challenging (high-variability, boundary) samples improves ACC in both i.i.d. and OOD scenarios, and reduces the forgetting rate in both i.i.d. and OOD scenarios.
>
> The table is as follows:
>
> \begin{array}{c|c|c|ccc|c|ccc|} \hline
> \textbf{Buffer} & & \textbf{500} & & \textbf{500} & & \textbf{1000} & & \textbf{1000} & \\\\ \hline
> \textbf{Balanced Class/Task} & & - & & \checkmark & & - & & \checkmark & \\\\ \hline
> \textbf{Memory Update Policy} & & \textbf{Reservoir} & \textbf{Random} & \textbf{Hard} & \textbf{Challenging} & \textbf{Reservoir} & \textbf{Random} & \textbf{Hard} & \textbf{Challenging} \\\\ \hline
> \textbf{ACC} & \textbf{i.i.d.} & 23.34 & \underline{28.12} & 14.23 & \textbf{29.27} & 33.03 & \underline{33.97} & 18.49 & \textbf{36.06} \\\\
> & \textbf{OOD} & 11.61 & \underline{13.65} & 7.20 & \textbf{14.81} & 14.98 & \underline{16.11} & 8.75 & \textbf{17.86} \\\\ \hline
> \textbf{BWT} & \textbf{i.i.d.} & -58.22 & \underline{-47.61} & -65.47 & \textbf{-41.78} & -48.70 & \underline{-36.98} & -57.61 & \textbf{-35.08} \\\\
> & \textbf{OOD} & -34.38 & \underline{-23.86} & -33.04 & \textbf{-18.38} & -31.35  & \underline{-17.07} & -29.51 & \textbf{-15.44} \\\\ \hline
> \end{array}
>
>
> \begin{array}{c|c|c|ccc|c|ccc|} \hline
> \textbf{Buffer} & & \textbf{2000} & & \textbf{2000} & & \textbf{Mean} & & \textbf{Mean} & \\\\ \hline
> \textbf{Balanced Class/Task} & & - & & \checkmark & & - & & \checkmark & \\\\ \hline
> \textbf{Memory Update Policy} & & \textbf{Reservoir} & \textbf{Random} & \textbf{Hard} & \textbf{Challenging} & \textbf{Reservoir} & \textbf{Random} & \textbf{Hard} & \textbf{Challenging} \\\\ \hline
> \textbf{ACC} & \textbf{i.i.d.} & 37.63 & \underline{40.89} & 23.42 & \textbf{42.24} & 31.33 & \underline{34.33} & 18.71 & \textbf{35.86} \\\\
> & \textbf{OOD} & 17.02 & \underline{18.95} & 11.01 & \textbf{20.57} & 14.54 & \underline{16.24} & 8.99 & \textbf{17.75} \\\\ \hline
> \textbf{BWT} & \textbf{i.i.d.} & -49.98 & \underline{-27.73} & -53.13 & \textbf{-24.69} & -52.30 & \underline{-37.44} & -58.74 & \textbf{-33.85} \\\\
> & \textbf{OOD} & -35.31 & \underline{-12.05} & -26.91 & \textbf{-7.85} & -33.68 & \underline{-17.66} & -29.82 & \textbf{-13.89} \\\\ \hline
> \end{array}

---

> > ### Author Response · Authors · 2024-11-25
> > **Integration of our memory update policy with ER**
> >
> > To further validate the generalizability of our buffer update policy, we integrated it into the ER method, replacing its random sampling strategy. The results, shown below for a buffer size of 1000 on the Split CIFAR-100 dataset, reaffirm that our policy enhances performance regardless of the loss function used.
> >
> > \begin{array}{c|c|c|ccc}
> > \hline
> > \textbf{Balanced Class/Task} & & - & & \checkmark \\\\ \hline
> > \textbf{Memory Update Policy} & & \textbf{Reservoir} & \textbf{Random} & \textbf{Hard} & \textbf{Challenging} \\\\ \hline
> > \textbf{ACC} & \textbf{i.i.d.} & 18.58 & \underline{19.30} & 12.49 & \textbf{20.44} \\\\
> > & \textbf{OOD} & 3.36 & \underline{3.42} & 2.78 & \textbf{3.51} \\\\ \hline
> > \textbf{BWT} & \textbf{i.i.d.} & -77.19 & \underline{-73.69} & -81.20 & \textbf{-72.14} \\\\
> > & \textbf{OOD} & -21.21 & \underline{-20.20} & -21.22 & \textbf{-19.99} \\\\ \hline
> > \end{array}
> >
> > As shown, our buffer update policy improves ACC and reduces forgetting (BWT) in both i.i.d. and OOD scenarios, even when integrated with the ER method. This demonstrates its effectiveness as a plug-and-play component for buffer-based methods using random update policies, beyond the context of our PCL loss.
> > We hope this additional analysis addresses your concerns and highlights the robustness and utility of our proposed buffer update mechanism.
> >
> > In response to your question about integrating another method with our proxy-based contrastive learning loss, we are currently performing the necessary experiments and will share the results in our next reply within three hours.

---

> ### Author Response · Authors · 2024-11-25
> **Answer to Q2 (Our PCL loss gain)**
>
> In response to your question about integrating another method with our proxy-based contrastive learning loss, we have combined SOIF [3] with our proxy-based contrastive loss and present the results in the following table. For comparison, we have also included the results of SOIF from Table 1 and Table 2. SOIF leverages second-order influences to make more informed decisions about which samples to retain in the buffer, using cross-entropy loss. To analyze the performance of our PCL loss, we replaced its loss with our proxy-based contrastive loss. We conducted experiments on the Split CIFAR-100 dataset with a buffer size of 500. The results are shown below:
>
>
>
>
> \begin{array}{c|c|c|c|}
> \hline
> \textbf{Method} & & \text{SOIF} & \text{SOIF}\_{+\mathbf{\mathcal{L}}_{\mathbf{\text{PCL}}}} \\\\
> \hline
> \textbf{ACC} & \textbf{i.i.d.} & 18.85 & \textbf{21.46} \\\\
> & \textbf{OOD} & 3.18 & \textbf{10.69}  \\\\
> \hline
> \textbf{BWT} & \textbf{i.i.d.} & -71.47 & \textbf{-69.24} \\\\
> & \textbf{OOD} & \textbf{-19.72} & -43.17 \\\\
> \hline
> \end{array}
>
>
> As evident across all metrics and scenarios (except BWT on OOD), we outperform the baseline. The better BWT (OOD) of the baseline, as discussed in our paper, indicates that it has not acquired sufficient knowledge to retain, let alone forget.
>
>
>
> **References**
>
>
> [1] De Lange, Matthias and Tuytelaars, Tinne. "Continual Prototype Evolution: Learning Online From Non-Stationary Data Streams", ICCV,2021.
>
>
> [2] Lucas Caccia, Rahaf Aljundi, Nader Asadi, Tinne Tuytelaars, Joelle Pineau, Eugene Belilovsky. "New Insights on Reducing Abrupt Representation Change in Online Continual Learning", ICLR,2022.
>
>
> [3] Zhicheng Sun, Yadong Mu, and Gang Hua. “Regularizing second-order influences for continual learning”, CVPR, 2023.

---

> > ### Comment · Reviewer_M5Un · 2024-11-26
> >
> > Thank you for the clarification. However, I still have a concern about the novelty of your ACR Loss. In paper [1], different types of proxy-based contrastive replay have already been discussed. Additionally, in paper [2], it is demonstrated that a simple proxy-based contrastive loss is equivalent to a cross-entropy (CE) loss. For this reason, I will not change my score.
> >
> >
> > [1] Huiwei Lin, Baoquan Zhang, Shanshan Feng*, Xutao Li, Yunming Ye. "PCR: Proxy-based Contrastive Replay for Online Class-Incremental Continual Learning" , CVPR2023
> > [2]  Lucas Caccia, Rahaf Aljundi, Nader Asadi, Tinne Tuytelaars, Joelle Pineau, Eugene Belilovsky. "New Insights on Reducing Abrupt Representation Change in Online Continual Learning", ICLR,2022.

---

> > > ### Author Response · Authors · 2024-11-28
> > >
> > > **Proxy-based Methods.**
> > >
> > > Experience Replay (ER) uses CE loss. It uses a memory buffer to store samples from older classes, replaying them alongside new-class samples during training. In this method, categorical probabilities are calculated in the same way for all classes. This setup gives older-class samples an advantage during gradient updates—they receive more positive gradients, while new-class samples face more negative ones. Over time, as more classes are added, the number of samples per class in the buffer shrinks, weakening the gradients for older classes.
> > >
> > > Separated Softmax Incremental Learning (SS-IL) addresses this by calculating separate categorical probabilities for old and new classes using two independent Softmax layers. This method cuts off gradient flow between old and new classes, preventing interference. However, this isolation creates a new challenge: the model struggles to distinguish old and new classes because the lack of shared gradients limits its ability to learn cross-class relationships.
> > >
> > > ER-ACE takes a different approach by introducing an asymmetric cross-entropy loss. This method calculates new-class probabilities similarly to SS-IL but handles old-class probabilities like ER. It selectively uses only certain anchor-to-proxy pairs for learning new classes. ER-ACE blocks the gradient flow from new-class learning to old-class proxies while maintaining the reverse flow. This helps preserve older-class knowledge but comes at the expense of new-class performance, as the model becomes biased toward older classes.
> > >
> > > ---
> > >
> > > **Contrastive-based Methods.**
> > >
> > > SCR uses a contrastive loss as an alternative for online class-incremental continual learning (CICL). This method calculates similarities between anchor-sample pairs within a batch, using all batch samples except the anchor and samples with the same label as the anchor. Unlike proxy-based methods, which depend on the number of classes, contrastive loss depends on the number of samples in a batch. While this flexibility is an advantage, it also limits effectiveness when memory buffers or batch sizes are small, leading to subpar performance with fewer replay samples.
> > >
> > > ---
> > >
> > > **Summary.**
> > >
> > > 1. **Gradient Imbalance Drives Forgetting**: A major cause of catastrophic forgetting is the imbalance in gradient flow between old and new classes. New-class gradients dominate, making new-class samples easier to distinguish while old-class samples become less separable. Managing this gradient imbalance is crucial for reducing forgetting.
> > >
> > > 2. **Limitations of Proxy-based Methods (such as CE loss)**: Proxy-based approaches control gradient flow by selecting specific anchor-to-proxy pairs for training. While this helps alleviate forgetting, it can harm the model’s ability to learn new classes effectively.
> > >
> > > 3. **Limitations of Contrastive-based Methods**: Contrastive approaches rely on anchor-to-sample pairs within the same batch and don’t use proxies, making them overly dependent on batch composition. While they lack flexibility, their selection strategy offers valuable insights for designing anchor-to-proxy pair selection.
> > >
> > > ---
> > >
> > > **Ours.**
> > >
> > > With these conclusions in mind, we propose a hybrid solution that combines the strengths of proxy-based and contrastive-based methods. Unlike approaches, that add anchor-to-sample pairs to anchor-to-proxy pairs in the loss function, we take a different route. Specifically, we replace sample-based pairs with proxies in the contrastive loss function.
> > >
> > > This modification changes how categorical probabilities are computed for each mini-batch, offering two major advantages. First, it speeds up convergence and improves robustness, allowing the model to handle limited samples better with the help of proxies. Second, the proxies are selected only from the classes present in the current training batch. This restricts gradient propagation to these classes.
> > >
> > > Rather than fully isolating gradients across all proxies, this approach allows gradients to flow when the corresponding classes appear in the same batch. As a result, old-class proxies, initially affected by negative gradients from new classes, can produce positive gradients that counteract forgetting. This dynamic enables the model to achieve better recognition of both old and new classes, outperforming existing methods in generalization and robustness.

---

### Official Review · Reviewer_RFy4 · 2024-11-04

**Soundness:** 2
**Presentation:** 2
**Contribution:** 2
**Rating:** 3
**Confidence:** 4

**Summary:**

The paper addressed the out-of-distribution generalization performance of rehearsal-based CL methods, which is a topic that has been overlooked by the community. The authors proposed Adaptive Contrastive Replay (ACR) which consists of two components: proxy-based contrastive learning and adaptive replay buffer management. The former encourages the model to maintain a clear class boundary while the latter populates the buffer with task- and class-balanced challenging samples to improve the model's stability and plasticity. Experiments on popular CL benchmarks showed improved performance with the proposed method with both iid and ood data.

**Strengths:**

1. The paper is well-motivated

2. The paper is generally easy-to-follow

**Weaknesses:**

There are some critical concerns about the proposed components, some missing explanations, and some room for improvement in the representations of the paper. Please refer to the Questions section.

**Questions:**

1. How does the proposed PCL loss differ from a standard cross entropy loss with temperature-scaled softmax?

The authors might elaborate more on this because I cannot see the difference between the proposed PCL loss (Eq. 4) and a stand softmax CE loss.

2. Is the confidence variance evolved to fixed after epoch E, and is this robust?

According to Algorithm 1, it seems that confidence is only recorded until epoch E (which is 5 in practice) whereas the total training lasts 50 epochs. Is that correct? If so, I am wondering how robust is this confidence variance. First, in the first 5 epochs, the model might not be able to converge to a stable state, which means that the predictions can be rather random. Is the confidence variance based on nearly random predictions reliable? I am not sure. Can the authors compare the confidence variance at the beginning of the task and at the end? Second, it seems reasonable that the model would refine by itself the decision boundary, which means that some challenging samples might become not challenging along training. If the confidence variance is fixed after epoch E, the the entire buffer seems to be static within a task, then the score cannot take this into account.

3. Are the challenging samples really helping?

The authors' claim to include challenging samples in the buffer is to maintain accurate decision boundaries for old tasks (lines 285-286), which should lead to less forgetting. However, in Table 4, the challenging column is not performing better than the random column in BWT, which suggests that the proposed selection mechanism might not be effective in maintaining the decision boundary of old tasks. It also raised the concern that the improvement of the proposed selection mechanism might just be a result of balanced sampling in the memory buffer, which is more or less well-known in the CL community.

4. Some explanations are missing

In lines 89-90, the implementation of the comparison is not explained, which makes Fig. 1 harder to understand

In lines 266-269, the claim that no task label needs to be stored is confusing. How are the samples arranged separately in the buffer (line 266) if no task label is known?

The concept of proxy is not clearly explained (lines 182-184)

---

> ### Author Response · Authors · 2024-11-25
> **Answer to Q1**
>
> Thank you for your insightful comment  We appreciate the opportunity to clarify your concerns. To address your concern regarding the difference between the proposed Proxy-based Contrastive Loss (PCL) and a standard cross-entropy loss with temperature-scaled softmax, we can focus on the following distinctions:
>
>
> **Key Differences:**
> 1. **Proxy-to-Sample Relationships vs. Sample-to-Proxy Relationships**:
>    - **PCL**: It explicitly models the relationship between samples and their corresponding proxies (representative class weights). Each sample interacts not only with its own proxy but also with proxies of all other classes, enabling the alignment of samples with their class prototypes while pushing them away from other class prototypes. This proxy-to-sample approach improves generalization by ensuring class separation and robustness to noise.
>    - **Standard Cross-Entropy with Temperature-Scaled Softmax**: It calculates the probability of each class for a sample using the dot product of the sample's embedding and class weights (proxies). The loss only optimizes the alignment of each sample to its true class, without directly incorporating explicit inter-class repulsion.
>
> 2. **Contrastive Learning Elements**:
>    - **PCL**: Inspired by contrastive learning, PCL focuses on expanding the pool of negative samples (all non-class proxies act as negatives) while also ensuring that positive alignments (with the class proxy) are strengthened. This approach is rooted in a proxy-based contrastive objective that draws samples closer to their proxies while distancing them from others.
>    - **Standard Cross-Entropy**: It does not explicitly include a mechanism for ensuring separation from non-class embeddings, which may lead to overlap in feature space, especially in high-dimensional settings.
>
> 3. **Handling Class Imbalance and Representation**:
>    - **PCL**: Designed to handle imbalanced datasets more effectively by operating at the proxy level. This ensures that rare classes or underrepresented tasks are given equal importance in learning robust boundaries.
>    - **Cross-Entropy**: It may suffer from overfitting to dominant classes in imbalanced settings because the optimization focuses directly on sample-level probabilities without leveraging the global class relationships.
>
> 4. **Computational Benefits**:
>    - **PCL**: Proxy-based computation reduces the need for detailed pairwise comparisons across all samples (a common drawback of standard contrastive loss). This makes it computationally efficient compared to full contrastive learning while maintaining the benefits of sample-to-proxy relationships.
>    - **Cross-Entropy with Temperature Scaling**: While computationally efficient, it lacks the ability to encode contrastive relationships or ensure well-separated class boundaries in the latent space.
>
>
>
> **Summary:**
> The proposed PCL loss fundamentally differs from standard cross-entropy loss with temperature-scaled softmax by integrating proxy-based contrastive learning principles. Specifically, PCL:
>
>    - **1.** Employs proxy-to-sample relationships to optimize embeddings for both alignment with class prototypes and separation from other class proxies.
>    - **2.** Improves robustness to noise and class imbalance by ensuring a structured representation space with explicit inter-class repulsion.
>    - **3.** Offers computational efficiency compared to full contrastive learning by avoiding pairwise sample comparisons while still leveraging a contrastive mechanism.
>
>
> These distinctions allow PCL to achieve better Out-of-Distribution (OOD) generalization and stability-plasticity trade-offs compared to standard cross-entropy loss, as evidenced by our experimental results in Section 4.2.

---

> > ### Comment · Reviewer_RFy4 · 2024-11-25
> > **About PCL loss**
> >
> > Thanks for the explanation about the PCL loss. To the best of my knowledge, the inter-class relation can also be modeled by CE loss with the temperatured softmax. I might not agree with the reason as here I quote: *The loss only optimizes the alignment of each sample to its true class, without directly incorporating explicit inter-class repulsion* to be the advantage of the proposed PCL loss.
> >
> > My question was probably too vague. Could you please explain by the formal definition (as in Eq.4) the difference between your proposed PCL and a cross-entropy loss with temperature-scaled softmax? This might help clarify the difference. For your reference, an example of the definition of cross-entropy loss with temperature-scaled softmax can the found in section 3.3 of the PCL paper [a].
> >
> > [a] Yao, Xufeng, et al. "Pcl: Proxy-based contrastive learning for domain generalization." Proceedings of the IEEE/CVF Conference on Computer Vision and Pattern Recognition. 2022.

---

> > > ### Author Response · Authors · 2024-11-25
> > > **PCL vs CE**
> > >
> > > The primary distinction between Proxy-Based Contrastive Loss (PCL) and Standard Cross-Entropy with Temperature-Scaled Softmax lies in the definition of \\( C \\), which directly influences the **negative pool**, the role of proxies, and the learning dynamics. In Standard Cross-Entropy, \\( C \\) encompasses all class labels in the dataset, creating a static, global comparison pool where proxies represent entire classes. In contrast, PCL defines \\( C \\) as the class labels present in the current batch, resulting in a dynamic, batch-specific pool where proxies adapt to batch composition.
> > >
> > > This difference has significant consequences. **Standard Cross-Entropy** forms positive and negative pairs globally: the anchor compares against proxies of all dataset classes, ensuring comprehensive global comparisons but introducing inefficiencies and biases, particularly under class imbalance. The static negative pool dilutes gradients for underrepresented classes, leading to challenges like the misclassification of older classes in incremental learning. **PCL**, on the other hand, limits interactions to batch-relevant classes, forming positive pairs between proxies and batch samples of the same class, while negative pairs arise from other batch samples. This dynamic approach focuses on meaningful comparisons, ensuring faster convergence and reducing bias.

---

> > > > ### Comment · Reviewer_RFy4 · 2024-11-25
> > > >
> > > > Thanks for the clarification. I believe we are on the same page now. I might still have two more concerns:
> > > >
> > > > 1. If I am not mistaken, the proposed PCL loss can also be regarded as a masked CE loss with temperated softmax, where classes not appear in the current mini-batch will be masked. Then the idea of *proxy* is not significant anymore.
> > > >
> > > > 2. This idea of using batch-wise mask on the CE loss has been explored by ER-ACE for replay-based CL. I am afraid that this might not be a significant contribution as the author claimed.

---

> > > > > ### Author Response · Authors · 2024-11-26
> > > > >
> > > > > In response to Q1, increasing the pool of negative samples allows us to indirectly account for sample-to-sample relationships, functioning similarly to a contrastive loss rather than merely applying a basic cross-entropy mask. Additionally, we effectively address class imbalance issues and reduce the gradient influence from previous classes. This approach goes beyond a simple masking of cross-entropy. Finally, by leveraging a proxy we obtain faster convergence.
> > > > >
> > > > > Regarding Q2, let me clarify the difference between us, ER-ACE, and ER-AML.

---

> > > > > > ### Comment · Reviewer_RFy4 · 2024-11-26
> > > > > >
> > > > > > Thanks for your clarification. Regarding your claim: *increasing the pool of negative samples allows us to indirectly account for sample-to-sample relationships*, it looks confusing to me. In your previous argument, if I am not mistaken, your loss only uses negatives from the current batch, instead of all classes. How would this result in an increased negative pool?
> > > > > >
> > > > > > My concern about the term *proxy* remains, as the role of proxy in the loss is not different from standard class prototypes/class weights in the linear output layer for a CE loss.
> > > > > >
> > > > > > Lastly, I still believe this proposed PCL loss is equivalent to a masked cross entropy with temperatured softmax I previously mentioned. The authors' explanation does not convince me.

---

> > > > > > > ### Author Response · Authors · 2024-11-28
> > > > > > >
> > > > > > > **Proxy-based Methods.**
> > > > > > >
> > > > > > > Experience Replay (ER) uses CE loss. It uses a memory buffer to store samples from older classes, replaying them alongside new-class samples during training. In this method, categorical probabilities are calculated in the same way for all classes. This setup gives older-class samples an advantage during gradient updates—they receive more positive gradients, while new-class samples face more negative ones. Over time, as more classes are added, the number of samples per class in the buffer shrinks, weakening the gradients for older classes.
> > > > > > >
> > > > > > > Separated Softmax Incremental Learning (SS-IL) addresses this by calculating separate categorical probabilities for old and new classes using two independent Softmax layers. This method cuts off gradient flow between old and new classes, preventing interference. However, this isolation creates a new challenge: the model struggles to distinguish old and new classes because the lack of shared gradients limits its ability to learn cross-class relationships.
> > > > > > >
> > > > > > > ER-ACE takes a different approach by introducing an asymmetric cross-entropy loss. This method calculates new-class probabilities similarly to SS-IL but handles old-class probabilities like ER. It selectively uses only certain anchor-to-proxy pairs for learning new classes. ER-ACE blocks the gradient flow from new-class learning to old-class proxies while maintaining the reverse flow. This helps preserve older-class knowledge but comes at the expense of new-class performance, as the model becomes biased toward older classes.
> > > > > > >
> > > > > > > ---
> > > > > > >
> > > > > > > **Contrastive-based Methods.**
> > > > > > >
> > > > > > > SCR uses a contrastive loss as an alternative for online class-incremental continual learning (CICL). This method calculates similarities between anchor-sample pairs within a batch, using all batch samples except the anchor and samples with the same label as the anchor. Unlike proxy-based methods, which depend on the number of classes, contrastive loss depends on the number of samples in a batch. While this flexibility is an advantage, it also limits effectiveness when memory buffers or batch sizes are small, leading to subpar performance with fewer replay samples.
> > > > > > >
> > > > > > > ---
> > > > > > >
> > > > > > > **Summary.**
> > > > > > >
> > > > > > > 1. **Gradient Imbalance Drives Forgetting**: A major cause of catastrophic forgetting is the imbalance in gradient flow between old and new classes. New-class gradients dominate, making new-class samples easier to distinguish while old-class samples become less separable. Managing this gradient imbalance is crucial for reducing forgetting.
> > > > > > >
> > > > > > > 2. **Limitations of Proxy-based Methods (such as CE loss)**: Proxy-based approaches control gradient flow by selecting specific anchor-to-proxy pairs for training. While this helps alleviate forgetting, it can harm the model’s ability to learn new classes effectively.
> > > > > > >
> > > > > > > 3. **Limitations of Contrastive-based Methods**: Contrastive approaches rely on anchor-to-sample pairs within the same batch and don’t use proxies, making them overly dependent on batch composition. While they lack flexibility, their selection strategy offers valuable insights for designing anchor-to-proxy pair selection.
> > > > > > >
> > > > > > > ---
> > > > > > >
> > > > > > > **Ours.**
> > > > > > >
> > > > > > > With these conclusions in mind, we propose a hybrid solution that combines the strengths of proxy-based and contrastive-based methods. Unlike approaches, that add anchor-to-sample pairs to anchor-to-proxy pairs in the loss function, we take a different route. Specifically, we replace sample-based pairs with proxies in the contrastive loss function.
> > > > > > >
> > > > > > > This modification changes how categorical probabilities are computed for each mini-batch, offering two major advantages. First, it speeds up convergence and improves robustness, allowing the model to handle limited samples better with the help of proxies. Second, the proxies are selected only from the classes present in the current training batch. This restricts gradient propagation to these classes.
> > > > > > >
> > > > > > > Rather than fully isolating gradients across all proxies, this approach allows gradients to flow when the corresponding classes appear in the same batch. As a result, old-class proxies, initially affected by negative gradients from new classes, can produce positive gradients that counteract forgetting. This dynamic enables the model to achieve better recognition of both old and new classes, outperforming existing methods in generalization and robustness.

---

> > > > > > > ### Author Response · Authors · 2024-11-29
> > > > > > >
> > > > > > > In cross-entropy (CE), each sample's focus is limited to its relationship with the **proxies of non-target classes**. In contrast, our method expands this perspective by considering the relationship between each sample and the **proxies of other samples' classes**, effectively enlarging the negative pool. This approach combines the strengths of sample-to-proxy interactions (similar to CE) and sample-to-sample relationships (as seen in contrastive loss), thereby bridging the gap between the two paradigms.

---

> > > > > ### Author Response · Authors · 2024-11-26
> > > > >
> > > > > Below, we present a detailed comparison of our approach with ER-AML and ER-ACE:
> > > > >
> > > > > ### **ACR Loss: Proxy-Based Contrastive Loss**
> > > > > - **Formulation**: Aligns the latent embedding \\( f_\theta(x) \\) (where \\( f_\\theta \\) is the encoder with parameters \\(\\theta\\), mapping the input \\(x\\) into the latent space) of a sample \\(x\\) with its class proxy \\(W_y\\) (a representative vector for the target class \\(y\\)) while separating it from proxies \\(W_L\\) of all other classes in the batch.
> > > > >   \\[
> > > > >   L_{\text{PCL}} = - \frac{1}{N} \sum_{i=1}^N \log \frac{\exp(f_\theta(x) \cdot W_y / \tau)}{\sum_{L \in C} \exp(f_\theta(x) \cdot W_L / \tau)}
> > > > >   \\]
> > > > >   - **\\(N\\)**: The total number of samples in the batch.
> > > > >   - **\\(\\tau\\)**: A temperature parameter controlling the sharpness of similarity distributions.
> > > > >   - **\\(C\\)**: The set of all class labels in the current batch.
> > > > >
> > > > > - **Objective**: Ensures compact clustering of embeddings for the same class and separation between clusters of different classes.
> > > > > - **Mechanism**:
> > > > >   - Optimizes the proxy-to-sample relationship for efficiency, reducing computational overhead by avoiding explicit pairwise comparisons between individual samples.
> > > > >   - Expands the pool of negatives by considering all non-target class proxies \\(W_L\\) as negatives.
> > > > > - **Key Focus**: Efficient and stable feature alignment using proxies.
> > > > >
> > > > > ---
> > > > >
> > > > > ### **ER-AML Loss: Asymmetric Metric Learning Loss**
> > > > > - **Formulation**: Combines two components:
> > > > >   1. **Supervised Contrastive Loss (SupCon)** for incoming data (\\(X_\text{in}\\), the set of new samples being learned):
> > > > >      \\[
> > > > >      L_1(X_\text{in}) = - \sum_{x_i \in X_\text{in}} \frac{1}{|P(x_i)|} \sum_{x_p \in P(x_i)} \log \frac{\text{sim}(f_\theta(x_p), f_\theta(x_i))}{\sum_{x_n \in N \cup P(x_i)} \text{sim}(f_\theta(x_n), f_\theta(x_i))}
> > > > >      \\]
> > > > >      - **\\(P(x_i)\\)**: The set of positive samples (other samples of the same class as \\(x_i\\) in the batch or buffer).
> > > > >      - **\\(N\\)**: The set of negative samples (samples from other classes in the batch or buffer).
> > > > >      - **\\(\\text{sim}(a, b)\\)**: The similarity between embeddings \\(a\\) and \\(b\\), calculated as:
> > > > >        \\[
> > > > >        \text{sim}(a, b) = \exp\left(\frac{a^\top b}{\tau \||a\|| \||b\||}\right)
> > > > >        \\]
> > > > >   2. **Cross-Entropy Loss** for replayed data (\\(X_\\text{bf}\\), the set of samples replayed from the buffer):
> > > > >      \\[
> > > > >      L_2(X_\text{bf}) = - \sum_{x \in X_\text{bf}} \log \frac{\text{sim}(W_{c(x)}, f_\theta(x))}{\sum_{c \in C_\text{all}} \text{sim}(W_c, f_\theta(x))}
> > > > >      \\]
> > > > >      - **\\(W_{c(x)}\\)**: The class proxy of the label \\(c(x)\\) corresponding to sample \\(x\\).
> > > > >      - **\\(C_\\text{all}\\)**: The set of all class labels observed so far.
> > > > >
> > > > > - **Combined Loss**:
> > > > >   \\[
> > > > >   L(X_\text{in} \cup X_\text{bf}) = \gamma L_1(X_\text{in}) + L_2(X_\text{bf})
> > > > >   \\]
> > > > >   - **\\(\\gamma\\)**: A weighting factor balancing the two terms.
> > > > >
> > > > > - **Objective**: Aligns embeddings of current task data while consolidating knowledge across previous tasks.
> > > > > - **Mechanism**:
> > > > >   - Uses pairwise relationships among samples (via \\(P(x_i)\\) and \\(N\\)) for finer control over embeddings.
> > > > >   - Incorporates class prototypes for replayed data to maintain stability.
> > > > >
> > > > > ---
> > > > >
> > > > > ### **ER-ACE Loss: Asymmetric Cross-Entropy Loss**
> > > > > - **Formulation**: Combines two cross-entropy losses:
> > > > >   \\[
> > > > >   L_{\text{ER-ACE}} = L_{\text{ce}}(X_\text{in}, C_\text{curr}) + L_{\text{ce}}(X_\text{bf}, C_\text{old} \cup C_\text{curr})
> > > > >   \\]
> > > > >   - **\\(C_\\text{curr}\\)**: The set of classes in the incoming batch.
> > > > >   - **\\(C_\\text{old}\\)**: The set of previously seen classes stored in the buffer.
> > > > >   - **\\(L_{\\text{ce}}(X, C)\\)**: The cross-entropy loss for samples \\(X\\) over classes \\(C\\):
> > > > >     \\[
> > > > >     L_{\text{ce}}(X, C) = - \sum_{x \in X} \log \frac{\exp(\text{sim}(W_y, f_\theta(x)))}{\sum_{c \in C} \exp(\text{sim}(W_c, f_\theta(x)))}
> > > > >     \\]
> > > > >
> > > > > - **Mechanism**:
> > > > >   - For \\(X_\\text{in}\\): Computes logits for only current classes \\(C_\\text{curr}\\), isolating new class learning.
> > > > >   - For \\(X_\\text{bf}\\): Includes both current and past classes (\\(C_\\text{curr} \\cup C_\\text{old}\\)), consolidating previous knowledge.
> > > > >
> > > > > - **Objective**: Indirectly enforces contrast by isolating logits for incoming data and leveraging replayed data for broader class alignment.
> > > > >
> > > > > ---

---

> > > > > ### Author Response · Authors · 2024-11-26
> > > > >
> > > > > ### **Key Differences Across Losses**
> > > > > 1. **Contrastive Mechanism**:
> > > > >    - ACR explicitly aligns embeddings with proxies (\\(W_y\\)), optimizing proxy-based relationships.
> > > > >    - ER-AML uses pairwise sample relationships, leveraging \\(P(x_i)\\) and \\(N\\).
> > > > >    - ER-ACE induces contrast implicitly via the asymmetric treatment of logits.
> > > > >
> > > > > 2. **Negative Sample Selection**:
> > > > >    - ACR: Considers all non-target proxies as negatives, broadening coverage.
> > > > >    - ER-AML: Relies on sample-based negatives within the batch or buffer.
> > > > >    - ER-ACE: Limits logits for negatives to the class sets \\(C_\\text{curr}\\) and \\(C_\\text{old}\\).
> > > > >
> > > > > 3. **Optimization Complexity**:
> > > > >    - ACR: Lower due to proxy-based design.
> > > > >    - ER-AML: Higher due to pairwise sample-to-sample comparisons.
> > > > >    - ER-ACE: Moderate, as it focuses on logits rather than explicit pairwise comparisons.

---

> ### Author Response · Authors · 2024-11-25
> **Answer to Q2**
>
> Thank you for highlighting this important question about the robustness of confidence variance and the implications of limiting its calculation to the first \\(E\\) epochs.
>
>
> To identify the most informative samples of each class, we utilized a data-centric artificial intelligence approach. In data-centric AI, the emphasis is on improving datasets to facilitate effective model training with fewer data requirements [1, 2]. For instance, as cited in our paper, [3] and [4] leverage model predictions during training to refine datasets. These studies demonstrate that early-stage uncertainty is a reliable indicator of a sample’s contribution to the training process. By analyzing network predictions for individual samples during the initial training epochs, they identify noisy labels and outlier samples, aiding dataset optimization.
>
>
> As you noted, during the training process, the decision boundary evolves. The key question is: **Which boundary leads to better generalization?** One shaped by outliers or one influenced by genuinely challenging (most informative) samples?
>
>
> To address this, we analyzed CIFAR10 in batch training (training all 10 classes at once, resembling one task of Split CIFAR-100 under continual learning). We calculated the target confidence of each sample at every epoch and displayed the mean and variance of sample confidence across these epochs in the Figure linked below. In the left sub-figure, we consider the first \\(E\\) epochs \\(E = 5\\) as in our paper, and in the right sub-figure, we use 50 epochs (matching the epoch count per task in our paper).
>
> [Please click to view the Figure](https://anonymous.4open.science/r/Figures-EB2E/1.png)
>
> We categorized the samples observed during the first \\(E = 5\\) epochs into three groups:
>
>    - **Area 1.** **Simple Samples:** High confidence with low variance, consistently well-classified and at the center of the distribution.
>
>    - **Area 2.** **Challenging Samples (Most Informative):** Moderate mean confidence with high variance, residing near the decision boundary. They are ideal for shaping a robust boundary.
>
>    - **Area 3.** **Hard (Outlier) Samples:** Low confidence with low variance, far from the decision boundary, offering limited value for boundary refinement.
>
> The bottom sub-figure shows representative images from these areas (Area 1, Area 2, and Area 3) for each CIFAR10 class. Each column represents a class, with rows corresponding to simple, challenging, and hard samples, respectively. Simple samples display clear class attributes (e.g., a horse on grass; grappling with the effects of spurious correlation). Challenging samples exhibit ambiguous traits (e.g., an airplane against a gray backdrop; enhancing OOD generalization), while hard samples feature unusual or rare characteristics (e.g., a horse’s head only), often functioning as outliers.
>
> As training progresses from \\(E = 5\\) to \\(E = 50\\):
>
>    - **Simple Samples** retain high confidence and low variance, remaining at the center of the distribution.
>
>    - **Challenging Samples** show increasing confidence (mean confidence rises) and decreasing variance as the model learns them robustly. Most migrate from the boundary to the center, while a few persist near the boundary.
>
>    - **Hard Samples** exhibit higher confidence but also increased variance, eventually achieving moderate confidence and defining the boundary.
>
> Thus:
>
>    - **Area 4** (\\(E = 50\\)) contains simple samples and most challenging samples.
>
>    - **Area 5** (\\(E = 50\\)) contains hard samples and some challenging samples, shaping the boundary.
>
> Comparing Area 2 (\\(E = 5\\)) and Area 5 (\\(E = 50\\)), we argue that Area 2, with challenging samples, provides better generalization. By prioritizing these **challenging (most informative) samples** for buffer selection, the decision boundary is shaped by meaningful boundary samples rather than outlier samples. This strategy "reminds" the model of an ideal boundary in future tasks, promoting generalization and stability-plasticity balance.
>
> In Figure 4 of the Appendix, we analyze the sensitivity of the hyperparameter \\(E\\). The results show stable performance across a range (\\(E = 2\\) to \\(7\\)) on all datasets, maintaining consistency in ACC and BWT under i.i.d. and OOD conditions. These findings confirm that identifying boundary samples requires only a few initial training epochs.
>
>
> Additionally, as shown in the Figure linked below, we conducted another experiment on CIFAR-100, extending the range of \\(E\\) from 2 to 14. The results demonstrate that performance gradually decreases beyond \\(E = 7\\). This is because outliers transition to challenging samples, and challenging samples to simple ones, reducing their impact on refining the decision boundary.
>
>
>
> [Please click to view the Figure](https://anonymous.4open.science/r/Figures-EB2E/2.png)

---

> ### Author Response · Authors · 2024-11-25
> **Answer to Q3**
>
> In Table 4, we initially demonstrated various buffer update policies, including “Reservoir,” “Hard,” “Challenging,” and “Random.” However, a discrepancy in the experimental setup led to the “Random” policy using augmented versions of samples in the buffer, whereas the other policies stored original (non-augmented) samples. This setup inadvertently favored the “Random” policy, resulting in better BWT compared to our proposed “Challenging” policy. We understand how this might raise concerns about the effectiveness of our method in maintaining decision boundaries for old tasks.
> To address this, we conducted a more equitable comparison by ensuring all policies stored original (non-augmented) samples in the buffer. Furthermore, while the original Table 4 was limited to a buffer size of 500 on the Split CIFAR-100 dataset, we extended the evaluation to include buffer sizes of 1000 and 2000. The updated results, presented in the table below, also include the mean performance across all buffer sizes.
>
> \begin{array}{c|c|c|ccc|c|ccc|} \hline
> \textbf{Buffer} & & \textbf{500} & & \textbf{500} & & \textbf{1000} & & \textbf{1000} & \\\\ \hline
> \textbf{Balanced Class/Task} & & - & & \checkmark & & - & & \checkmark & \\\\ \hline
> \textbf{Memory Update Policy} & & \textbf{Reservoir} & \textbf{Random} & \textbf{Hard} & \textbf{Challenging} & \textbf{Reservoir} & \textbf{Random} & \textbf{Hard} & \textbf{Challenging} \\\\ \hline
> \textbf{ACC} & \textbf{i.i.d.} & 23.34 & \underline{28.12} & 14.23 & \textbf{29.27} & 33.03 & \underline{33.97} & 18.49 & \textbf{36.06} \\\\
> & \textbf{OOD} & 11.61 & \underline{13.65} & 7.20 & \textbf{14.81} & 14.98 & \underline{16.11} & 8.75 & \textbf{17.86} \\\\ \hline
> \textbf{BWT} & \textbf{i.i.d.} & -58.22 & \underline{-47.61} & -65.47 & \textbf{-41.78} & -48.70 & \underline{-36.98} & -57.61 & \textbf{-35.08} \\\\
> & \textbf{OOD} & -34.38 & \underline{-23.86} & -33.04 & \textbf{-18.38} & -31.35  & \underline{-17.07} & -29.51 & \textbf{-15.44} \\\\ \hline
> \end{array}
>
>
> \begin{array}{c|c|c|ccc|c|ccc|} \hline
> \textbf{Buffer} & & \textbf{2000} & & \textbf{2000} & & \textbf{Mean} & & \textbf{Mean} & \\\\ \hline
> \textbf{Balanced Class/Task} & & - & & \checkmark & & - & & \checkmark & \\\\ \hline
> \textbf{Memory Update Policy} & & \textbf{Reservoir} & \textbf{Random} & \textbf{Hard} & \textbf{Challenging} & \textbf{Reservoir} & \textbf{Random} & \textbf{Hard} & \textbf{Challenging} \\\\ \hline
> \textbf{ACC} & \textbf{i.i.d.} & 37.63 & \underline{40.89} & 23.42 & \textbf{42.24} & 31.33 & \underline{34.33} & 18.71 & \textbf{35.86} \\\\
> & \textbf{OOD} & 17.02 & \underline{18.95} & 11.01 & \textbf{20.57} & 14.54 & \underline{16.24} & 8.99 & \textbf{17.75} \\\\ \hline
> \textbf{BWT} & \textbf{i.i.d.} & -49.98 & \underline{-27.73} & -53.13 & \textbf{-24.69} & -52.30 & \underline{-37.44} & -58.74 & \textbf{-33.85} \\\\
> & \textbf{OOD} & -35.31 & \underline{-12.05} & -26.91 & \textbf{-7.85} & -33.68 & \underline{-17.66} & -29.82 & \textbf{-13.89} \\\\ \hline
> \end{array}
>
> The updated results demonstrate that across all buffer sizes, metrics (ACC, BWT), and scenarios (i.i.d., OOD), our “Challenging” policy consistently outperforms the “Random” policy, which achieves the second-best performance. This indicates that under fair experimental conditions, our policy is highly effective at capturing better decision boundaries.
> To further validate the generalizability of our buffer update policy, we integrated it into the ER method, replacing its random sampling strategy. The results, shown below for a buffer size of 1000 on the Split CIFAR-100 dataset, reaffirm that our policy enhances performance regardless of the loss function used.
>
> \begin{array}{c|c|c|ccc}
> \hline
> \textbf{Balanced Class/Task} & & - & & \checkmark \\\\ \hline
> \textbf{Memory Update Policy} & & \textbf{Reservoir} & \textbf{Random} & \textbf{Hard} & \textbf{Challenging} \\\\ \hline
> \textbf{ACC} & \textbf{i.i.d.} & 18.58 & \underline{19.30} & 12.49 & \textbf{20.44} \\\\
> & \textbf{OOD} & 3.36 & \underline{3.42} & 2.78 & \textbf{3.51} \\\\ \hline
> \textbf{BWT} & \textbf{i.i.d.} & -77.19 & \underline{-73.69} & -81.20 & \textbf{-72.14} \\\\
> & \textbf{OOD} & -21.21 & \underline{-20.20} & -21.22 & \textbf{-19.99} \\\\ \hline
> \end{array}
>
> As shown, our buffer update policy improves ACC and reduces forgetting (BWT) in both i.i.d. and OOD scenarios, even when integrated with the ER method. This demonstrates its effectiveness as a plug-and-play component for buffer-based methods using random update policies, beyond the context of our PCL loss.
> We hope this additional analysis addresses your concerns and highlights the robustness and utility of our proposed buffer update mechanism.

---

> ### Author Response · Authors · 2024-11-25
> **Answer to Q4 (Regarding lines 89-90)**
>
> Thank you for your comment. We acknowledge that the explanation of the comparison methodology could have been clearer in mentioned lines 89-90. The implementation details for the comparison are already included in the manuscript (**Section 4.1 SETUP**) but may not have been sufficiently emphasized in mentioned lines 89-90. To clarify:
>
> 1. **Setup for Comparison:**
>    All methods were implemented under identical conditions using the Mammoth framework as described in [5] and [6]. The training was conducted under class-incremental settings, as done in previous studies [6], on the Split CIFAR-100, Split Mini-ImageNet, and Split Tiny-ImageNet datasets, using buffer sizes of 500, 1000, and 2000. Detailed training conditions and hyperparameters were outlined in the **Datasets** and **Implementation Details** paragraphs. Results were averaged over five independent runs to ensure fairness.
>
> 2. **OOD Sample Generation (Paragraph: Datasets):**
>    OOD samples were generated following the methodology of [7]. This involved applying standard corruptions, such as Gaussian Noise, Motion Blur, Elastic Transform, and so on, to the test datasets. These corruptions simulate real-world distributional shifts, providing a robust assessment of OOD generalization.
>
> 3. **Evaluation Metrics (Paragraph: Metrics):**
>    Average Accuracy (ACC) and Backward Transfer (BWT), defined in Eq. 9, were used to evaluate performance. ACC measures overall task performance, while BWT quantifies the extent of forgetting. These metrics, crucial for assessing i.i.d. and OOD scenarios, were discussed in detail in the **Metrics** Paragraph.
>
> 4. **Findings in Fig. 1:**
>    Fig. 1 shows the average ACC over three buffer sizes in the OOD scenario for each dataset. The results highlight the substantial performance degradation of existing methods on OOD samples compared to i.i.d. scenarios.
>
> To improve clarity, we will enhance the manuscript by cross-referencing these paragraphs and adding some of the above explanations directly to lines 89–90. We hope this addresses the concern and improves the reader's understanding of the methodology. Thank you again for your valuable feedback.

---

> ### Author Response · Authors · 2024-11-25
> **Answer to Q4 (Regarding lines 266-269)**
>
> Thank you for bringing up this important point. The mechanism for organizing samples in the buffer without explicitly storing task labels is based on the structure of our proposed buffer. We understand that this may not have been adequately clarified in the manuscript. Allow us to explain:
>
> 1. **Sequential Task Arrangement:**
>    The buffer is updated sequentially as tasks are encountered. This means that samples from earlier tasks naturally occupy the lower indices in the buffer, while samples from later tasks occupy higher indices. Thus, the task order is implicitly preserved in the buffer structure without requiring explicit task labels.
>
> 2. **Class-Specific Organization:**
>    Within each task, the buffer is further organized by class. Samples belonging to the same class are stored together and ordered by their confidence variance (\\( \\sigma^2 \\)). This eliminates the need to store task labels or \\( \\sigma^2 \\) explicitly because the buffer structure inherently reflects the task and class arrangement.
>
> 3. **Efficient Access Without Task Labels:**
>    To identify task-specific samples, the buffer leverages its index-based organization. Samples are accessed based on their position in the buffer, which corresponds to their task and class membership due to the sequential and class-based storage scheme.
>
> To improve the clarity of this explanation, we will revise the corresponding lines in the manuscript to explicitly describe how the buffer organization allows task and class separation without storing task labels. Thank you for pointing out this opportunity for enhancement.

---

> ### Author Response · Authors · 2024-11-25
> **Answer to Q4 (Regarding lines 182-184)**
>
> Thank you for pointing out the need to clarify the concept of proxies in our method. The explanation on lines 182–184 may not have fully conveyed the rationale behind their use or their specific implementation in our approach. We provide the following clarification:
>
> 1. **Definition of Proxy:**
>    In our method, proxies are representative weights that correspond to specific classes. Each proxy acts as a learned anchor point in the embedding space for its associated class, summarizing the class's characteristics. Instead of comparing all individual sample pairs, we use these proxies to establish relationships between classes and samples in the batch, significantly reducing computational complexity.
>
> 2. **Role in Proxy-Based Contrastive Learning:**
>    The proxies are utilized in the contrastive learning objective to align each sample's embedding with its class proxy while distancing it from other class proxies. This is achieved through the proxy-based contrastive loss, defined in Eq. (4). This approach enables robust representation learning by focusing on proxy-to-sample relationships rather than exhaustive sample-to-sample comparisons, which can be computationally prohibitive.
>
> 3. **Advantages of Using Proxies:**
>    - **Efficiency:** By reducing the need for pairwise comparisons, proxies enable faster and more stable convergence.
>    - **Noise Tolerance:** Proxies are more robust to noisy or outlier samples, as they summarize the class characteristics rather than relying on individual samples.
>    - **Generalization:** Proxies provide a smoother embedding space by focusing on class-level relationships.
>
> We will revise the explanation in the manuscript (lines 182–184) to explicitly define proxies and their role in our approach. Additionally, we will enhance the connection between proxies and their impact on the computational and generalization benefits in Section 3.2. Thank you for this constructive feedback, which will help us improve the clarity of our manuscript.
>
>
>
> **References**
>
>
> [1] Mohammad Motamedi, Nikolay Sakharnykh, and Tim Kaldewey. A data-centric approach for training deep neural networks with less data. arXiv preprint arXiv:2110.03613, 2021.
>
> [2] Mark Mazumder, Colby Banbury, Xiaozhe Yao, Bojan Karlas, William Gaviria Rojas, Sudnya Diamos, Greg Diamos, Lynn He, Alicia Parrish, Hannah Rose Kirk, et al. Dataperf: Benchmarks for data-centric ai development. arXiv preprint arXiv:2207.10062, 2022.
>
> [3] Mariya Toneva, Alessandro Sordoni, Remi Tachet des Combes, Adam Trischler, Yoshua Bengio, and Geoffrey J Gordon. An empirical study of example forgetting during deep neural network learning. arXiv preprint arXiv:1812.05159, 2018.
>
> [4] Swabha Swayamdipta, Roy Schwartz, Nicholas Lourie, Yizhong Wang, Hannaneh Hajishirzi, Noah A Smith, and Yejin Choi. Dataset cartography: Mapping and diagnosing datasets with training dynamics. arXiv preprint arXiv:2009.10795, 2020.
>
> [5] Pietro Buzzega, Matteo Boschini, Angelo Porrello, Davide Abati, and Simone Calderara. Dark experience for general continual learning: a strong, simple baseline. Advances in neural information processing systems, 33:15920–15930, 2020.
>
> [6] Matteo Boschini, Lorenzo Bonicelli, Pietro Buzzega, Angelo Porrello, and Simone Calderara. Classincremental continual learning into the extended der-verse. IEEE transactions on pattern analysis and machine intelligence, 45(5):5497–5512, 2022.
>
> [7] Dan Hendrycks and Thomas Dietterich. Benchmarking neural network robustness to common corruptions and perturbations. arXiv preprint arXiv:1903.12261, 2019.

---

### Official Review · Reviewer_uazR · 2024-11-04

**Soundness:** 2
**Presentation:** 3
**Contribution:** 2
**Rating:** 3
**Confidence:** 4

**Summary:**

This paper proposes Adaptive Contrastive Replay (ACR) to address the Out-of-Distribution (OOD) generalization problem often overlooked by existing class-incremental learning algorithms. While previous research has achieved strong performance across various class-incremental learning scenarios, these algorithms tend to perform well only on in-domain tasks due to issues with bad memorization, resulting in poor generalization to OOD tasks. To overcome this, the paper introduces a Proxy-based Contrastive Loss and Adaptive Replay Buffer Management. The Proxy-based Contrastive Loss utilizes the class weights as proxies for contrastive learning. Additionally, exemplars are sampled based on the variance of confidence scores, allowing for balanced sampling across tasks and classes and prioritizing hard examples close to decision boundaries to improve stability and OOD generalization. Through extensive experiments and analysis, the proposed algorithm demonstrates superior performance across diverse scenarios.

**Strengths:**

1. This paper identifies the often-overlooked OOD generalization issue in class-incremental learning and proposes ACR to address it. I find the distinction between bad and good memorization for explaining this problem, along with the design of Proxy-based Contrastive Loss and Adaptive Replay Buffer Management, to be well-motivated and effectively structured to tackle the identified issue.

2. To validate the proposed algorithm, the authors conducted extensive experiments across various datasets, algorithms, and memory buffer sizes. Additionally, the thorough analysis and ablation studies to verify and examine the effectiveness of the algorithm are insightful and engaging to read.

**Weaknesses:**

1. The paper [1] proposes a sampling method that stores the most interfered samples in the replay memory. I believe this plays a similar role and function described in equation (3) of the author's paper. Therefore, I think it is necessary for the authors to explain the differences from [1] and consider it as an additional baseline.

2. The authors validated the superiority of their proposed algorithm through various experiments using CIFAR-100, Split Mini-ImageNet, and Split Tiny-ImageNet in online class-incremental learning. However, I have the following questions regarding these experimental results:

   2-1) According to the paper, the results for Split CIFAR-100 in Table 1 show the experimental results for 10 tasks using the ResNet-18 model. I understand that reporting results for these 10 tasks have generally been done with the ResNet-32 model, starting with [2], and results have been reported in papers such as [3], [4], and [5]. When comparing these results (especially Figure 1(a) of [4]) with those in Table 1, the results in this paper appear to be relatively low. For example, when using a replay memory size of 2000 in [4] and [5], the accuracy of the naive algorithm ER (or replay) is reported to be around 40%, while Table 1 shows only 24%. Furthermore, the highest performance reported in [4] is 60% (e.g., DER), whereas the proposed algorithm achieved only 42% in Table 4. Is this discrepancy solely due to the model used? I believe that comparing the performance of the proposed algorithm with existing baselines using ResNet-32 is essential to confirm its superiority.

   2-2) Since all experiments were conducted using images sized 32 x 32 for a specific scenario, it makes it very difficult to assess whether the proposed algorithm can achieve excellent results across diverse scenarios. Additionally, many class-incremental learning algorithms are known to exhibit different performance trends depending on the input image size and scenario (shown in [4] and [5]), so I believe the authors should conduct additional experiments using ImageNet (or not resized ImageNet-100) and other task scenarios (e.g., a scenario in which a large number of classes are learned in the first task, and the remaining classes are divided and learned in subsequent tasks.) to demonstrate the effectiveness of their proposed algorithm in various settings.

3. From the results in Table 4, it is evident that the simple 'Random' policy already performs very close to 'Challenging' (ACR). This experiment was conducted with a replay memory size of 500; what would happen if it were increased to 2000? Would 'Challenging' still outperform 'Random' in this case? I think it is crucial to demonstrate that 'Challenging' provides superior performance compared to 'Random' in this experiment.

4. Additionally, I believe that, for the ablation study on Adaptive Replay Buffer Management, it is necessary to include results from experiments using ER without proxy-based contrastive learning in Table 4. This will allow for a proper evaluation of the standalone effectiveness of Adaptive Replay Buffer Management.

[1] Online Continual Learning with Maximally Interfered Retrieval, NeurIPS 2019.
[2] iCaRL: Incremental Classifier and Representation Learning, CVPR 2017
[3] GDumb: A Simple Approach that Questions Our Progress in Continual Learning, ECCV 2020
[4] PyCIL: A Python Toolbox for Class-Incremental Learning, Arxiv.
[5] Rebalancing Batch Normalization for Exemplar-based Class-Incremental Learning, CVPR 2023

**Questions:**

I have included all weaknesses and questions in the Weakness section, so please refer to it. I strongly resonate with the authors' concerns about the OOD generalization problem in class-incremental learning algorithms, and I believe the proposed algorithm is well-designed. However, the various questions raised by the experiments and the lack of sufficient ablation studies make it difficult to provide a more favorable evaluation.I look forward to the authors correcting any misunderstandings I may have and providing a response to my review in their author response.

---

> ### Author Response · Authors · 2024-11-20
> **Answer to Q1**
>
> Thank you for your insightful comment. We appreciate the opportunity to clarify the differences between our approach and the MIR method [1].
>
> In continual learning, buffer management involves two critical phases: buffer update and buffer retrieval. The buffer update phase determines which samples from previous tasks should be stored in the buffer, while the buffer retrieval phase selects samples from the buffer to train alongside current task data.
>
> Our approach specifically focuses on the buffer update phase: we populate the buffer with the most variable samples from previous classes. For buffer retrieval during training, we use a random sampling policy, akin to Experience Replay (ER) [6]. This means we do not introduce any special selection strategy for retrieving samples during training.
>
> In contrast, the MIR method [1] adopts a different focus. MIR does not apply a specific strategy for buffer updates and uses reservoir sampling to populate the buffer, similar to ER [1] [7]. However, MIR implements a targeted strategy for buffer retrieval by selecting samples that are most interfered with—i.e., those whose predictions are most negatively impacted by expected parameter updates. This retrieval mechanism aims to prioritize replaying samples that are most vulnerable to interference from new tasks. As discussed in [7], selecting samples based on a significant increase in loss can lead to redundancy since similar samples in the latent space may all exhibit increased loss, potentially reducing diversity in the training data.
>
> The following statements from the MIR paper [1] support our claim:
> “In this work, we consider a controlled sampling of memories for replay.”
> “We restrict ourselves to the use of reservoir sampling for deciding which samples to store.”
>
> In summary, our approach focuses on buffer update rather than buffer retrieval, which is the opposite of the approach in MIR. Additionally, our method and MIR differ in how they select informative samples: we prioritize high-variance samples, while MIR selects samples based on the increase in loss projected from the anticipated parameter updates with new data. As illustrated in [7], [8], [9], and [10] the performance of MIR on Split Mini-ImageNet and Split CIFAR-100 is nearly close to the ER method, highlighting that their targeted retrieval strategy does not always lead to significant improvements over random sampling.

---

> ### Author Response · Authors · 2024-11-20
> **Answer to Q2-1**
>
> Thank you for pointing this out. We appreciate the opportunity to clarify the discrepancies you highlighted regarding the results in Table 1 compared to those reported in [4] and [5].
>
> As mentioned in lines 292 and 348, for a fair comparison, we modified the Mammoth framework as outlined in [11, 12], similar to how MetaSP and SOIF [9, 10] modified it. By following the SOIF and MetaSP, our results are directly comparable to theirs. For example, references [9, 10] for Experience Replay (ER) on Split CIFAR-100 with a buffer size of 1000 achieved an ACC of 17.56%, while we achieved 18.58%. Moreover, MetaSP [9] and SOIF [10] for their corresponding methods on Split CIFAR-100 with a buffer size of 1000 reported ACC of 25.72% and 27.99%, respectively. We reported ACC of 25.15% and 26.19% for them, respectively. This indicates that accuracy differs across frameworks based on certain elements, and in the Mammoth framework, our results fall within this expected range. Meanwhile, several factors influence the reported performance, resulting in lower accuracy compared to [4] and [5]. So we would like to elaborate on some of these factors:
>
> Different backbone networks, hyperparameters (e.g., number of epochs, memory batch size), and validation metrics can significantly affect accuracy. Following [9, 10, 11, 12], we used ResNet-18 (line 349), set the number of epochs per task to 50 (line 353), a memory batch size of 32 (line 351), and averaged the ACC over five fixed seeds (line 350). We applied standard data augmentations including random crop and random horizontal flip (mentioned in line 355).
>
> In [4] (as evident in the full version of [4] and their corresponding GitHub code; [13]), they set the number of training epochs to 170, used a memory batch size of 128, and utilized ResNet-32 as the backbone network. Moreover, they reported the Top-1 accuracy and used additional techniques like herding and class balance for the buffer, along with color jitter in addition to random crop and horizontal flip.
>
> In [5], their naive method differs from the standard ER. They updated the buffer after each task with balanced class random sampling, employed normalization techniques, set the number of epochs per task to 160, and used ResNet-32 as the backbone network.
>
> The standard ER uses reservoir sampling, where the buffer is updated as each batch arrives with random samples, leading to potential class imbalances. Also, ER uses random retrieval.
>
> Given these differences, it is understandable that our reported accuracy is lower than those reported in [4] and [5] for both ER and our method compared to DER.
>
> We hope this explanation helps clarify the differences in accuracy between our work and the baselines mentioned. We believe that despite these discrepancies, our modifications to the Mammoth framework provide a fair comparison with related methods, and our results demonstrate competitive performance within the context of our chosen settings. However, if you feel that including results on ResNet-32 is essential, we are open to conducting the necessary experiments and reporting them in the camera-ready version, though this will require significant time.

---

> ### Author Response · Authors · 2024-11-20
> **Answer to Q2-2**
>
> Regarding your concern about the image sizes, we have conducted experiments using the original 84×84 image size of Split Mini-ImageNet without resizing, in addition to resizing the images to 32×32. Due to computational constraints, we were unable to perform experiments on the full ImageNet dataset or the non-resized ImageNet-100. Consequently, we stay focused on Split Mini-ImageNet to address your concern about image sizes. Additionally, due to the limited time available during the rebuttal phase, these experiments have been conducted with a buffer size of 500 and seed 0. If this is insufficient, we are prepared to conduct experiments across all buffer sizes and average the results over five fixed seeds (0 to 4) as done in the paper and report them in the camera-ready version.
> The experimental results are summarized below:
>
>
>
> \begin{array}{c|c|c|cccccccc}
> \hline
> & \textbf{Method} & & \textbf{GEM} & \textbf{A-GEM} & \textbf{ER} & \textbf{GSS} & \textbf{GDUMB} & \textbf{HAL} & \textbf{MetaSP} & \textbf{SOIF} & \textbf{Ours} \\\\
> \hline
>  & \textbf{ACC} & \textbf{i.i.d.}  & 17.60 & 14.71 & 16.64 & 18.77 & 7.17 & 4.80 & \underline{19.70} & 18.33 & \textbf{26.19} \\\\
> \boldsymbol{32 \times 32}& \textbf{ACC}  & \textbf{OOD} & 2.93 & 2.65 & 3.23 & 3.33 & 1.42 & 2.38 & \underline{3.35} & 3.20 & \textbf{11.97} \\\\
> & \textbf{BWT} & \textbf{i.i.d.} & -52.49 & -66.79 & -67.04 & -62.00 & - & \textbf{-38.79} & -65.34 & -65.46 & \underline{-46.30} \\\\
> & \textbf{BWT} & \textbf{OOD} & \textbf{-9.85} & -12.85 & -12.91 & -11.59 & - & -11.31 & \underline{-11.03} & -12.81 & -23.80 \\\\
> \hline \hline
>  & \textbf{ACC} & \textbf{i.i.d.}  & \underline{25.36} & 16.34 & 19.53 & 20.97 & 8.8 & 7.61 & 24.59 & 20.37 & \textbf{33.45} \\\\
> \boldsymbol{84 \times 84} & \textbf{ACC} & \textbf{OOD} & 3.92 & 3.41 & 3.74 & 3.43 & 1.73 & 2.97 & \underline{4.16} & 3.37 & \textbf{17.38} \\\\
> & \textbf{BWT} & \textbf{i.i.d.}  & -64.59 & -73.14 & -76.74 & -71.86 & - & \underline{-45.1} & -69.91 & -75.35 & \textbf{-34.65} \\\\
> & \textbf{BWT} & \textbf{OOD} & -11.59 & -13.03 & -13.01 & -12.55 & - & \underline{-11.46} & \textbf{-10.83} & -13.34 & -23.91 \\\\
> \hline
> \end{array}
>
> The data show that increasing the image size not only does not degrade performance but improves our algorithm's advantage. For the 32×32 images, we outperform the second-best on ACC (i.i.d.) and ACC (OOD) by margins of 6.49% and 8.62%, respectively. Regarding BWT (i.i.d.) (32×32), the HAL method has the lowest ACC (i.i.d.) (4.80%), leading to the best BWT (i.i.d.) due to insufficient learning (i.e., nothing to forget). Our method achieves the second-best BWT (i.i.d.), and for a fair comparison, BWT should be considered alongside ACC (as discussed in Section 4.2 of the paper). For example, compared to GEM, which has a high ACC (i.i.d.) (17.60%) and the third-best BWT (i.i.d.), we outperform by a margin of 6.19%. Also, for MetaSP, which has the second-best ACC (i.i.d.) (19.70%), we outperform it by a margin of 19.16% in BWT (i.i.d.). In the OOD scenario, the ACC of all baselines is too low, resulting in better BWT due to insufficient knowledge capture.
>
> When increasing the image size from 32×32 to 84×84 (original Mini-ImageNet size), our margin of improvement increases further. For ACC (i.i.d.) and ACC (OOD), we outperform the second-best by margins of 8.09% and 13.22%, respectively. Regarding BWT (i.i.d.), we outperform the second-best by 10.45%. Also, for GEM, which has the second-best ACC (i.i.d.) (25.36%) and the third-best BWT (i.i.d.), we outperform it by a margin of 29.94% in BWT (i.i.d.). Again, for the BWT (OOD) case, the ACC for all methods is too low, indicating that they have not acquired sufficient knowledge to retain, let alone forget. This raises concerns about their OOD generalization capabilities.
>
> Regarding your concern about other task scenarios, as mentioned in the full version [13] of the referenced paper [4], there are two different ways to split the classes into incremental stages: 1. Train from Scratch (TFS) and 2. Train from Half (TFH). In [13], the authors state, "Both of these settings are widely adopted in the current CIL community.” As evident in Table 5 and Table 6 in the appendix of the full version of the paper [4], for almost all methods, the ACC for TFS is lower than the ACC for TFH. This is because having more stages of learning introduces greater challenges to retaining knowledge from earlier classes. In contrast, TFH presents a less severe challenge, as the model primarily fine-tunes its knowledge base to accommodate smaller increments of new classes after establishing a large initial set. Furthermore, since we followed the Mammoth framework, which uses the TFS approach, we adopted the same. However, if this remains a concern, we can prepare the results for TFH for the camera-ready version.

---

> ### Author Response · Authors · 2024-11-20
> **Answer to Q3**
>
> To address your concern about whether increasing the buffer size from 500 to 2000 affects the performance of our method (‘Challenging’) compared to the ‘Random’ policy, we have repeated the experiments in Table 4 with a buffer size of 2000. The updated results are as follows:
>
>
> \begin{array}{c|c|c|ccc|c|ccc|}
> \hline
> \textbf{Buffer} & & \textbf{500} & & \textbf{500} & & \textbf{2000} & &  \textbf{2000}  & \\\\
> \hline
> \textbf{Balanced Class/Task} & & - &  & \checkmark &  & - &  & \checkmark & \\\\
> \hline
> \textbf{Memory Update Policy} & & \textbf{Reservoir} & \textbf{Random} & \textbf{Hard} & \textbf{Challenging} & \textbf{Reservoir} & \textbf{Random} & \textbf{Hard} & \textbf{Challenging} \\\\
> \hline
> \textbf{ACC} & \textbf{i.i.d.} & 23.34 & \underline{28.12} & 14.23 & \textbf{29.27} & 37.63 & \underline{40.89} & 23.42 & \textbf{42.24}  \\\\
> & \textbf{OOD} & 11.61 & \underline{13.65} & 7.20 & \textbf{14.81} & 17.02 & \underline{18.95} & 11.01 & \textbf{20.57} \\\\
> \hline
> \textbf{BWT} & \textbf{i.i.d.} & -58.22 & \underline{-47.61} & -65.47 & \textbf{-41.78} & -49.98 & \underline{-27.73} & -53.13 & \textbf{-24.69} \\\\
> & \textbf{OOD} & -34.38 & \underline{-23.86} & -33.04 & \textbf{-18.38} & -35.31 & \underline{-12.05} & -26.91 & \textbf{-7.85} \\\\
> \hline
> \end{array}
>
>
> From these results, it is evident that our method (‘Challenging’) continues to outperform the ‘Random’ policy across all metrics, even with the increased buffer size. Thank you for pointing out the importance of evaluating the effect of a larger buffer size on the performance comparison between the 'Challenging' and 'Random' policies. This additional analysis strengthens the robustness of our proposed approach and confirms that the superiority of ‘Challenging’ is not limited to smaller buffer sizes. We hope this addresses your concern thoroughly.

---

> ### Author Response · Authors · 2024-11-20
> **Answer to Q4**
>
> Thank you for your insightful comment regarding the need to evaluate the standalone effectiveness of Adaptive Replay Buffer Management without proxy-based contrastive learning.
>
> In response, we have repeated the experiments presented in Table 4 by incorporating our adaptive replay buffer management into ER, excluding proxy-based contrastive learning. The results are as follows:
>
> \begin{array}{c|c|c|ccc}
> \hline
>  \textbf{Balanced Class/Task} &  & - & & \checkmark \\\\ \hline
> \textbf{Memory Update Policy} &  & \textbf{Reservoir} & \textbf{Random} & \textbf{Hard} & \textbf{Challenging} \\\\
> \hline
> \textbf{ACC} & \textbf{i.i.d.} & 18.58 & \underline{19.30} & 12.49 & \textbf{20.44} \\\\
> & \textbf{OOD} & 3.36 & \underline{3.42} & 2.78 & \textbf{3.51} \\\\
> \hline
> \textbf{BWT} & \textbf{i.i.d.} & -77.19 & \underline{-73.69} & -81.20 & \textbf{-72.14} \\\\
> & \textbf{OOD} & -21.21 & \underline{-20.20} & -21.22 & \textbf{-19.99} \\\\ \hline
> \end{array}
>
> These experiments were conducted on Split CIFAR-100 with a buffer size of 1000. As shown, our buffer management consistently enhances the performance of ER across all metrics, demonstrating the effectiveness of our adaptive update policy.
>
>
> **References**
>
> [1] Online Continual Learning with Maximally Interfered Retrieval, NeurIPS 2019.
>
> [2] iCaRL: Incremental Classifier and Representation Learning, CVPR 2017
>
> [3] GDumb: A Simple Approach that Questions Our Progress in Continual Learning, ECCV 2020
>
> [4] PyCIL: A Python Toolbox for Class-Incremental Learning, Arxiv.
>
> [5] Rebalancing Batch Normalization for Exemplar-based Class-Incremental Learning, CVPR 2023
>
> [6] On tiny episodic memories in continual learning. arXiv preprint arXiv:1902.10486, 2019.
>
> [7] Online Class-Incremental Continual Learning with Adversarial Shapley Value, AAAI 2021.
>
> [8] Supervised Contrastive Replay: Revisiting the Nearest Class Mean Classifier in Online Class-Incremental Continual Learning, CVPR 2021
>
> [9] Exploring example influence in continual learning, NeurIPS 2022.
>
> [10] Regularizing second-order influences for continual learning, CVPR 2023.
>
> [11] Dark experience for general continual learning: a strong, simple baseline, NeurIPS, 2020.
>
> [12] Classincremental continual learning into the extended der-verse, PAMI 2022.
>
> [13] Class-Incremental Learning: A Survey, Da-Wei Zhou, Qi-Wei Wang, Zhi-Hong Qi, Han-Jia Ye, De-Chuan Zhan, Ziwei Liu.

---

> ### Author Response · Authors · 2024-11-23
> **A friendly reminder**
>
> We are grateful for your valuable comments and hope that our responses have addressed your concerns. If you have any further concerns, please let us know. We look forward to hearing from you.

---

> ### Author Response · Authors · 2024-11-25
> **A kind reminder**
>
> We appreciate your reviews. We hope that our responses have adequately addressed your concerns. As the deadline for open discussion nears, we kindly remind you to share any additional feedback you may have. We are keen to engage in further discussion.

---

> > ### Comment · Reviewer_uazR · 2024-11-25
> > **Thank you for the responses!**
> >
> > First of all, I would like to express my gratitude to the authors for their detailed and thorough responses to my review. I also sincerely apologize for the delay in my reply, as I have been preoccupied with other commitments. I have carefully reviewed the authors' responses as well as the reviews provided by other reviewers, and my feedback is as follows:
> >
> > 1. Answer to Q1
> >
> > Thank you for providing a clear comparison. I believe that including a summary of this explanation in the updated version of the paper would enhance the clarity of the paper’s contribution.
> >
> > 2. Answer to Q2-1
> >
> > Thank you for elaborating on the differences between the two implementations. I now understand that this paper's implementation is based on the Mammoth framework and that the authors have faithfully reported the performance of existing algorithms using this framework. Additionally, I recognize that the performance differences arise due to differences in implementation details, such as the model, epoch settings, and augmentations, as mentioned in my review.
> >
> > I agree that differences in hyperparameters like epochs and mini-batch size are not major concerns, as each algorithm has its own set of optimal hyperparameters. However, I still have the following questions regarding the performance differences related to the choice of the model:
> >
> > 2.1) ResNet-18 has approximately 11M parameters, while ResNet-32 has only around 0.4M parameters. Despite the significantly larger parameter count of ResNet-18, the results reported in the paper show lower performance compared to ResNet-32. I suspect this is because ResNet-32 is a model specifically adapted for CIFAR datasets, making it more suitable for this dataset. This observation suggests that ResNet-32 is better aligned with experiments involving CIFAR-100. From this perspective, the results using ResNet-18 seem less relevant and may not carry as much significance.
> >
> > For experiments involving standard ResNet models, I believe it would be more appropriate to conduct experiments on datasets with an ImageNet scale (e.g., ImageNet-1k or at least ImageNet-100). Additionally, to address this concern, I recommend including results using ResNet-32 on CIFAR-100 for comparison.
> >
> > 3. Answer to Q2-2
> >
> > Thank you for the detailed response.
> >
> > 3.1) The improved results with 84×84 images are encouraging. However, the Split mini-ImageNet dataset is designed for few-shot learning, and its overall scale is not very large. As such, I believe, as mentioned in 2.1, that presenting results on ImageNet-100 would help address these concerns more comprehensively.
> >
> > 3.2) While TFS is indeed a more challenging setting than TFH, I find TFH to be more practical from an application standpoint. Evaluating algorithms under both settings provides valuable perspectives on their performance in diverse scenarios. Many papers referenced in my review consider both settings, and I believe including experimental results for both would make the paper’s contributions more compelling.
> >
> > 4. Answer to Q3 and Q4
> >
> > Thank you for sharing the additional experimental results. The results for larger buffer sizes and the ablation study are promising. Including these results in the main manuscript would enhance the paper’s persuasiveness.
> >
> > Summary
> >
> > Once again, I apologize for the delayed response. The authors’ replies have addressed some of my concerns. However, I still question whether the experimental settings, particularly those involving CIFAR-100 with ResNet-18, are entirely appropriate (see 2.1). This uncertainty raises doubts about whether the proposed algorithm would demonstrate similar superiority when tested on datasets and models that are better aligned.
> >
> > I believe resolving these concerns through further experiments during the limited discussion period may not be feasible. Taking this into account, my current rating for the paper is 4. However, since the system only allows a choice between 3 and 5, I am maintaining my current rating of 3. I remain open to additional discussions with the authors or other reviewers.

---

### Meta-Review · Area_Chair_AGUd · 2024-12-23

**Metareview:**

The paper proposes Adaptive Contrastive Replay (ACR), which adaptively updates the replay buffer while employing a proxy-based contrastive loss for online class-incremental learning tasks. The paper is well-written and experimental results show some promising results. However, the proposed ACR loss shares some similarity with previously proposed methods, and a more fine-grained explanation should be given to make a clear distinction among them. While the authors gave lengthy explanation during the rebuttal period, incorporating all those points into the final version would require a significant restructuring of the paper. Therefore, the current decision is Reject, but AC encourages the authors to incorporate all the results/explanation that the authors gave in the rebuttal phase to the updated manuscript and resubmit in a future venue.

**Additional Comments On Reviewer Discussion:**

Authors provided a very lengthy rebuttal and there were active engagements.

uazR raised several important points regarding the main contribution of the paper. Some of the comments on requiring more experimental results seemed to be a little too much, but most of them resulted in helping authors to clarify their main points. Authors provided some more results for clarification.

RFy4 persistently raised the point on the comparison of ACR loss with previously proposed similar loss functions. Despite authors explanation, the reviewer was not fully convinced.

---

### Decision · Program_Chairs · 2025-01-22

Reject